

# Climate variability, heat distribution and polar amplification in the unipolar 'doubthouse' of the Oligocene

Dominique K.L.L. Jenny[1], Tammo Reichgelt[2], Charlotte L. O'Brien[3], Xiaoqing Liu[4], Peter K. Bijl[1], Matthew Huber[4], Appy Sluijs[1]

[1] Department of Earth Sciences, Utrecht University, 3584 CB, Netherlands
[2] Department of Earth Sciences, University of Connecticut, CT 06269-1045, USA
[3] Department of Geography, University College London, WC1E 6BT, UK
[4] Department of Earth, Atmospheric, and Planetary Sciences, Purdue University, IN 47907, USA

10    *Correspondence to*: Dominique K.L.L. Jenny (d.k.l.l.jenny@uu.nl)



**Abstract**

The Oligocene (33.9 – 23.03 Ma) was characterised by generally warm climates, with flattened meridional temperature gradients while Antarctica retained a significant cryosphere. This makes the Oligocene an imperfect analogue to long-term future climate states with unipolar icehouse conditions. Although local and regional climate and environmental reconstructions of Oligocene conditions are available, the community lacks synthesis of regional reconstructions. In order to provide a comprehensive overview of marine and terrestrial climate and environmental conditions in the Oligocene, as well as a reconstruction of trends through time, we here review marine and terrestrial proxy records and numerical climate model simulations of the Oligocene. Results display weaker temperature gradients during the Oligocene compared to modern times, with generally warm poles and colder-than-modern temperatures around the equator. Sea surface temperatures (SSTs) show similar trends to the land temperatures, with warm temperatures around the mid and high latitudes (∼60 – 90°) of especially the Southern Hemisphere. Vegetation-based precipitation reconstructions of the Oligocene suggest drier conditions compared to modern times, in particular around the equator. When compared to proxy-based data, climate modelling approaches overestimate Oligocene precipitation in most areas, in particular the tropics. Temperature around the mid to high latitudes is generally underestimated in models compared to proxy data and tend to overestimate the warming in the tropics. In line with previous conclusions models underestimate polar amplification and the equator-to-pole heat distribution that prevailed during the Oligocene. Despite prevalent glaciation on Antarctica, the Oligocene "icehouse" experienced warm global average temperatures while still maintaining a unipolar icehouse state.



## 1 Introduction

Simulations of future climate change, by current generation fully coupled climate models, indicate that global average surface warming will continue over the coming centuries depending on future $CO_2$ emissions and sequestration (IPCC, 2022). The models, as well as available temperature, $CO_2$ and sea level reconstructions of past Mesozoic and Cenozoic warm climates, suggest that Earth's climate may ultimately equilibrate towards unipolar conditions, with ice only remaining on Antarctica (Burke et al., 2018; Clark et al., 2016). Climate models additionally predict a global equilibrium surface warming between 1.5–4.5°C per doubling of atmospheric $CO_2$ concentrations relative to pre-industrial values, with a most likely value around 3–4°C (IPCC, 2022). This warming will be amplified at higher latitudes, notably the Arctic, by a factor of 2–3 relative to the global average (Fischer et al., 2018; Holland and Bitz, 2003; IPCC, 2022). However, model projections, particularly for such distant future non-analogue states, still include large uncertainties and are ideally independently constrained by data. Proxy-based reconstructions of past climates provide useful insights into the Earth's natural response to $CO_2$ changes and therefore are an independent opportunity to quantify the sensitivity of various climate parameters to greenhouse forcing, including sea-level and polar amplification (e.g., Burke et al., 2018; Lunt et al., 2016; Palaeosens Project Members, 2012). This way, climate models simulating past climate conditions can be compared against proxy data, and thus their accuracy can be evaluated.

It is likely that important climate parameters such as equilibrium climate sensitivity and polar amplification depend on the state of the climate (e.g., Farnsworth et al., 2019; Gaskell et al., 2022; Hutchinson et al., 2021; Köhler et al., 2015; Masson-Delmotte et al., 2013). Therefore, these parameters have been investigated for several past climate states. Traditional targets include the Pleistocene, Pliocene and Eocene (e.g., Burke et al., 2018) and, more recently, the Miocene (Steinthorsdottir et al., 2021). These time intervals encompass a wide range of climate states, including those with ice sheets in both the Southern (SH) and Northern Hemisphere (NH), the Southern Hemisphere only, and ice-free states, in addition to a wide range of atmospheric greenhouse gas concentrations (e.g., Rae et al., 2021).

Recent work has highlighted the Oligocene as a potentially useful climate state to assess the dynamics of global climate under a climate state with only an Antarctic ice sheet (O'Brien et al., 2020). Though geographical boundary conditions during the Oligocene (33.9–23.03 million years ago (Ma)) were different to today, the Oligocene is the most recent analogue to future unipolar icehouse climate states (e.g., O'Brien et al., 2020; Liebrand et al., 2017; Miller et al., 1988). Sparse glacially deposited sediments suggest the presence of NH glaciers as young as the late Eocene (Eldrett et al., 2007; St. John, 2008), but there is no evidence for late Eocene large-scale continental glaciation. Instead, the cryosphere potentially comprised localized glaciers and restricted sea ice in the Arctic Ocean (DeConto et al., 2008; Stickley et al., 2009). Reconstructions of atmospheric $CO_2$ range from over 1000 parts per million (ppm) to as low as ~300 ppm for the Oligocene (Foster et al., 2017; Rae et al., 2021), similar to the range projected for the future based on various emission scenarios (IPCC, 2022). Despite potentially low atmospheric $CO_2$ conditions, the few available sea surface temperature (SST) reconstructions indicate warmer than modern climates throughout the Oligocene, with remarkable polar amplification (O'Brien et al., 2020).



Across the Eocene-Oligocene transition (EOT), atmospheric $CO_2$ concentrations dropped from >1000 ppm during the Eocene (56.0–33.9 Ma) to ~750 ppm or lower at the beginning of the Oligocene (Heureux and Rickaby, 2015; Pagani et al., 2005; Pearson et al., 2009). This drop coincides with a large increase ($\sim$1–1.5 ‰) in deep ocean benthic foraminifer oxygen isotope

ratios ($\delta^{18}O$), which includes the effects of both the formation of ice sheets and a drop in deep-sea temperatures (e.g., Coxall & Wilson, 2011). The forcings underlying the onset of the Oligocene so-called 'icehouse' climate (i.e., with polar ice) are still highly debated. As of now the leading hypothesis invokes a strongly non-linear response to orbital forcing superimposed on a long-term drop in atmospheric $CO_2$ levels across a critical threshold (DeConto et al., 2008; DeConto and Pollard, 2003; Galeotti et al., 2016). However, the question that remains is if, or to what extent, tectonic changes and associated oceanographic

reorganizations in the Southern Ocean (SO) played a role (e.g., Hill et al., 2013; Houben et al., 2019; Huber et al., 2004; Ladant et al., 2014; Sauermilch et al., 2021). Changes associated with the onset of polar glaciation include a drop in the global average temperature (Eldrett et al., 2009; Kotthoff et al., 2014; Liu et al., 2009; Meckler et al., 2022; Sluiter et al., 2022; Thompson et al., 2021; Zanazzi et al., 2007) and a profound change in deep-water temperatures (Meckler et al., 2022). Across the EOT, surface cooling (Liu et al., 2009), the accumulation of land ice that reached the Antarctic coastlines (Salamy and Zachos,

1999), and the consequent appearance of sea ice (Houben et al., 2013), were associated with pronounced changes in SH atmospheric circulation and oceanographic conditions (Diester-Haass & Zahn, 1996; Houben et al., 2019; Liu et al., 2009; Tripati et al., 2005) and an increase in the poleward ocean heat transport (Goldner et al., 2014). The drop in temperatures at the beginning of the Oligocene also influenced the global turnover of flora and fauna (e.g., Solé et al., 2020; Sun et al., 2014). Oceanographic changes, including upwelling and the formation of sea ice, rapidly transformed circum-Antarctic marine

ecosystems (e.g., Houben et al., 2013; Salamy and Zachos, 1999), which might have facilitated the evolution of odontocete and mysticete (baleen) whales (e.g., Fordyce, 1980; Salamy and Zachos, 1999). Thus, the EOT seems to mark a prominent change in the global climate system (Westerhold et al., 2020) with the expansion of continental ice sheets.





**Figure 1: a. In blue: SST of ODP 1168A (west of Tasmania, TEX86H, Guitián & Stoll, 2021; Hoem et al., 2021) and in brown: SST of U1404 (northwest Atlantic, Uk37, Liu et al., 2018) b. Published pCO2 records of the Oligocene, black line represents the smoothed trend. Green squares: phytoplankton data, brown triangles: leaf gas exchange reconstructions, blue dots: boron isotopic data (Greenop et al., 2019; Moraweck et al., 2019; Pagani et al., 2005, 2011; Roth-Nebelsick et al., 2014; Witkowski et al., 2018; Zhang et al., 2013) c, d. Deep ocean benthic foraminifera stable carbon isotope and oxygen isotope records, respectively (Westerhold et al.**



**2020, notably representing the record of Pälike et al., 2006). Colour block red: Eocene-Oligocene Tansition (EOT), grey: Eocene-Oligocene Glacial Minima (EOGM), blue: Middle Oligoene Glacial (MOG), green: Oligocene-Miocene Transition (OMT).**

Although the Oligocene has been the subject of numerous studies, the documentation of global Oligocene climate conditions and its variability, including the hemispheric distribution of heat, meridional temperature gradients, and biotic change is sparse. High-resolution benthic foraminifer $\delta^{18}O$ records suggest significant variability in continental ice volume, paced by eccentricity and obliquity (e.g., De Vleeschouwer et al., 2017; Galeotti et al., 2016; Levy et al., 2019; Liebrand et al., 2017; Naish et al., 2001; Pälike et al., 2006b). Multiple-proxy SST data, albeit of much lower resolution than the deep ocean $\delta^{18}O$ records, revealed that the Oligocene was characterised by generally warm climates, with flattened meridional temperature gradients (Gaskell et al., 2022; O'Brien et al., 2020). Still, Antarctica retained a significant cryosphere (e.g., Hoem et al., 2021). The recorded trends, cycles and events provide ample opportunity to study the dynamics of climate and the carbon cycle in what has been called a 'doubthouse' or 'intermediate' climate state (O'Brien et al., 2020). Here, we aim to review the current state of knowledge regarding the Oligocene climate. To this end, after a chronostratigraphic section, we provide basic constraints regarding important climatic boundary conditions, such as paleogeography and atmospheric $CO_2$ levels. Then we compile various types of climate proxy data, including a new compilation and analysis of terrestrial fossil plant assemblages to assess long-term trends and variability. Finally, we identify specific points of interest for follow up research.



## 2 Oligocene chronostratigraphy and event nomenclature

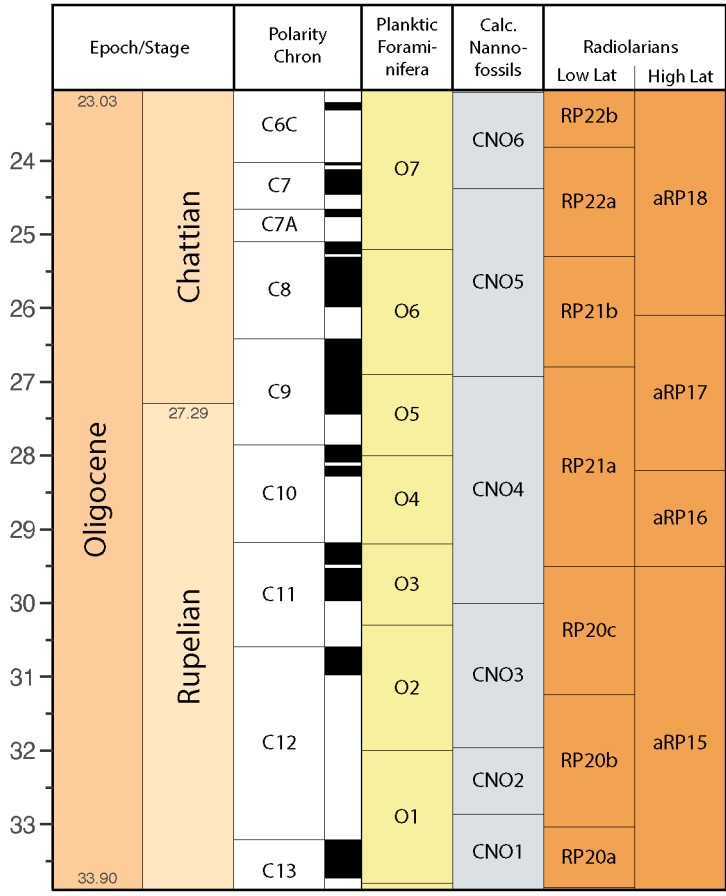

**Figure 2: Oligocene biostratigraphic zones. Planktic Foraminifera zones after Berggren et al., (2018), calcareous nannofossil zones by (Agnini et al., 2014). Radiolarian zonation for the low latitudes of the Pacific, Atlantic and Indian Ocean as defined by (Kamikuri et al., 2012) and for the high latitudes of the Southern Ocean from (Hollis et al., 2020). All given ages are on GTS2020 timescales (Gradstein et al., 2020).**

The Oligocene represents the epoch between two formal Global Stratotype Section and Points (GSSP), the Eocene-Oligocene Boundary (EOB) and the Oligocene-Miocene Boundary (OMB), at 33.9 Ma and 23.03 Ma following the GTS2020 time scale (Gradstein et al., 2020). The EOB GGSP was set in 1992, at the Massignano Quarry (Italy) and is defined by the extinction of the two planktic foraminifer genera *Hantkenina* and *Cribrohantkenina* at 33.9 Ma (Silva and Jenkins, 1993). The GSSP for the OMB was defined by Steininger et al., (1997) in the Piedmont Tertiary Basin in Italy on the magnetic reversal from polarity

chron C6Cn.2r–C6Cn.2n between two subunits of the Rigoroso Formation, which was later dated by Beddow et al., (2018) at 23.040 ±0.1 Ma. Within the Oligocene, Hardenbol & Berggren (1978) were the first to distinguish the Rupelian (33.9–27.29 Ma) from the Chattian (27.29–23.040 Ma) of northwestern Europe. They separated the two periods based on lithostratigraphy





in Belgium, into an open marine, clayey unit which overlies a shallower marine, sandy unit. The Rupelian (Chron C13r–C9n) was introduced by Dumont (1849), describing the Boom Clay Formation along the river Rupel and Scheldt in Belgium. The Chattian (Chron C9n–C6Cn) was officially first mentioned by Fuchs, (1894) who studied the "Kasseler Meeressande" (marine sands) in Hessen as well as Bünde, Germany (De Man et al., 2010; Van Simaeys, 2004; Van Simaeys et al., 2004). The GSSP for the Rupelian–Chattian boundary was set in 2016 by Coccioni et al., (2018) at the Monte Cagnero section near Urbania (Italy) and was bound by the (highest) last common occurrence (LCO) of the planktonic foraminifer *Chiloguembelina cubensis* at the base of the planktonic foraminifer zone O5. Currently, the official GSSP age for the Rupelian–Chattian boundary is 27.29 Ma, after Coccioni et al. (2018).

Several informal definitions are used to describe the various stratigraphic events associated with the Oligocene (Fig. 1). The Eocene-Oligocene transition (EOT) refers to the numerous climatic and environmental events broadly associated with the Epoch boundary (e.g., Coxall & Pearson, 2007; Eldrett et al., 2009; Houben et al., 2012; Zanazzi et al., 2007). However, it was recently defined as the ~790 kyr interval between the extinction of the coccolithophore species *Discoaster saipanensis* (~34.46 Ma) and the Earliest Oligocene Oxygen Isotope Step (EOIS) (Hutchinson et al., 2021). The EOIS — also known as the onset of Oligocene oxygen isotope zone 1 (Oi-1; Miller et al., 1991) is a ~40 kyr long lasting $\delta^{18}O$ shift ($\geq$0.7‰ increase; Coxall & Wilson, 2011; Zachos et al., 1996) which peaks at ~ 33.71 Ma. The EOT and the EOIS are followed by the Early Oligocene Glacial Maximum (EOGM), which lasted ~490 kyr from ~33.71 to ~33.22 Ma and chronostratigraphically correlates to most of the GPTS Magnetochron C13n (33.726–33.214 Ma) (Hutchinson et al., 2021). The EOGM was first introduced by Liu et al. (2004) and combines the separate $\delta^{18}O$ maxima which were defined by Zachos et al. (1996) as Oi-1a (~33.66 Ma) and Oi-1b (~33.26 Ma) as a consecutive period of colder climate/glaciation. Along with Oi-1, Miller et al. (1991) defined Oi-2 (30.3–25.1 Ma), which was later separated into Oi-2a (~28.3 Ma), Oi-2b (27.3–26.3 Ma, Chron C9n, NP24/NP25 boundary) and Oi-2c (~25.1 Ma) (Pekar et al., 2002, 2006; Pekar & Miller, 1996). The Oi-2b cooling event (Chron C9n, NP24/NP25 boundary) is also called the (Mid-) Oligocene Glacial Maximum/Interval (OMG/MOGI) and represents a ~1 Ma long phase of profound cooling/glacial expansion (Liebrand et al., 2017). The MOGI ended with three warming phases (~26.3, ~25.5 and ~24.22 Ma) leading up to the Oligocene-Miocene transition (OMT, 23.88–23.04 Ma). The beginning of the Miocene (23.040–5.33 Ma) is marked by the Mi-1 event (Miller et al., 1991; 23.883 Ma) during which deep ocean benthic foraminifer $\delta^{18}O$ values roughly increased by 1 ‰, followed by two rapid decreases of around 0.6 ‰ at ~23.64 and ~23.59 Ma (Billups et al., 2002; Flower et al., 1997; Miller et al., 1991).

## 2.1 Planktonic foraminifera zones

The first Oligocene planktonic foraminifera zones were defined by Blow (1979) who proposed the Paleocene zones P18–P21 to the Rupelian and P21–N4 (Neogene) to the Chattian. Wade et al. (2011), and later Berggren et al. (2018), proposed seven Oligocene (O1–O7) planktonic foraminifera zones building on work by (Berggren & Pearson (2005). O1, the '*Pseudohastigerina naguewichiensis* Highest-occurrence Zone', is defined by the interval between the Highest Occurrence





(HO) of *Hantkenina alabamensis* and *Pseudohastigerina naguewichiensis* and has an estimated age of around 33.8–32.0 Ma. The '*Turborotalia ampliapertura* Highest-occurrence Zone' is the second zone (O2, 32.0–30.3 Ma) and is defined by the interval between the HO of *Pseudohastigerina naguewichiensis* and the HO of *Turborotalia ampliapertura*. The third zone (O3) lasts from 30.3–29.2 Ma and is defined by the interval between the HO of *Turborotalia ampliapertura* and the Lowest Occurrence (LO) of *Globigerina angulisuturalis*. Zone O4 ranges from the LO of *Globigerina angulisuturalis* and the Highest

Common Occurrence (HCO) of *Chiloguembelina cubensis* and lasted from around 29.2–28.0 Ma. From 28.0–26.9 Ma lies the fifth zone (O5) which is between the HCO of *Chiloguembelina cubensis* and the HO of *Paragloborotalia opima.* The so called '*Globigerina ciperoensis* Partial-range Zone' (O6) ranges from 26.9–25.2 Ma and the HO of *Paragloborotalia opima* and the LO of *Paragloborotalia pseudokugleri*. The last zone, O7, lasts from 25.2–23.03 Ma and is defined by the interval between the LO of *Paragloborotalia pseudokugleri* and the LO of *Paragloborotalia kugleri*.

**2.2 Calcareous nannofossil zones**

The first calcareous nannofossil zones of the Oligocene were proposed by Martini (1971) and Okada & Bukry (1980). Martini (1971) proposed the Rupelian zones NP21–NP24 and the Chattian NP24–NP25. Okada & Bukry (1980) proposed a parallel zonation scheme with the zones CP16–CP19 to the Rupelian and CP19 to the Chattian. (Agnini et al., 2014) reassigned the Oligocene 6 calcareous nannofossil biozones (CNO1–CNO6). CNO1 is the so called '*Ericsonia formosa/Clausicocous*

*subdistichus* Concurrent Range Zone' and is defined by the base common occurrence (BCO) of *Clausicocous subdistichus* to the top occurrence (TO) of *Ericsonia formosa.* The zone lasts around 0.96 Ma, and the estimated age lies at 33.90–32.86 Ma. The zone corresponds to the upper part of zones of NP21 and CP16a/b. CNO2 is the '*Reticulofenestra umbilicus* Top Zone' and is defined from the TO of *Ericsonia formosa* to the TO of *Reticulofenestra umbilicus*. CNO2 corresponds to NP22 and CP16c and lasts from around 32.86 to 31.96 Ma. The third zone CNO3 lasts from 31.96 to 30.0 Ma and is called the

'*Dictyococcites bisectus* Partial Range Zone'. The CNO3 lasts from the TO of *Reticulofenestra umbilicus* to the base occurrence (BO) of *Sphenolithus distentus* and corresponds to the lower part of NP23 and to zone CP17. CNO4 is the '*Sphenolithus distentus*/*Sphenolithus predistentus* Concurrent Range Zone', includes the BO of *Sphenolithus distentus* and the TO of *Sphenolithus predistentus* and lasts from 30.0 to 26.93 Ma. It encompasses the upper part of NP23 and most of NP24 as well as CP18 and most of CP19a. The fifth zone CNO5, is defined by the interval from the TO *Sphenolithus predistentus*

to the TO of *Sphenolithus ciperoensis.* The so called '*Sphenolithus ciperoensis* Top Zone' (26.93–24.38 Ma) lies at the bottom of zone NP25 and includes the subzone CP19b. The last Oligocene zone CNO6 is the '*Triquetrorhabdulus carinatus* Partial Range Zone', and it lasts from 24.38 to 23.07 Ma and is defined by the TO of *Sphenolithus ciperoensis* to the TO of *Sphenolithus delphix.* CNO6 corresponds to the upper part of Zone NP25 and the lowermost of NN1 as well as CN1a and the lowermost part of CN1b.





## 2.3 Radiolarian biostratigraphic zones

Sanfilippo & Nigrini (1998) proposed a radiolarian biostratigraphic zones for the lower latitudes of the Pacific, Indian and Atlantic Oceans for the Oligocene (RP20–RP22). These low latitude zonations were revised by (Kamikuri et al., 2012) and 7 new subzones (RP20a–c, RP21a–b, RP22a–b) were proposed. RP20 is the *Theocyrtis tuberosa* Interval Zone (33.86–29.50 Ma), it is defined by the evolutionary transition (ET) of *Tristylospyris triceros* to the top occurrence of *Dorcadospyris ateuchus* and the ET from *Lithocyclia aristotelis* group to the base of *Lithocyclia angusta*. RP20a lasts from around 33.86–33.04 Ma and its end is marked the first occurrence (FO) of *Theocyrtis tuberosa.* RP20b starts with the first occurrence of *T. tuberosa* and is ended around 31.24 Ma by the FO of *Eucyrtidium plesiodiaphanes.* The third subzone (RP20c) lasts from 31.24 to 29.50 Ma and is ended with the transition from the last occurrence (LO) of *Tristylospyris triceros* to the FO of *Dorcadospyris ateuchus*. Zone RP21 is the *Dorcadospyris ateuchus* Interval Zone and lasts from 29.50 to 25.30 Ma. It started with the ET of *T. triceros* to *D. ateuchus* and is ended by the FO of *Lychnocanoma elongata.* It is divided into the two subzones RP21a (29.50–26.80 Ma) and RP21b (26.80–25.30 Ma) with RP21a ending/RP21b starting with the FO of *Lychnocanoma apodora.* The last Oligocene low latitude zone is RP22 (25.30–22.47 Ma) which is started with the FO of *L. elongata* and ended by the FO of *Cyrtocapsella tetrapera.* It is also split up into subzones RP22a (25.30–23.82 Ma) and RP22b (23.82–22.47 Ma) them being separated by the FO of *Eucyrtidium diaphanes* at the base of RP22b. For the mid and high latitudes of the Southern Ocean Hollis et al. (2020) revised some of the previously set Radiolarian zonations (Speijer et al., 2020). They came up with four new so called 'RPA zones' (aRP15–aRP18) for the Oligocene. aRP15 is the *Axoprunum? irregularis* zone and is defined by the LO of *Eucyrtidium antiquum* and it has its base at 33.65 Ma. There is also a secondary base suggested for the zone, which lies at 34.69 Ma and concurs with the LO of *Axoprunum? irregularis.* The base of aRP16 lies at 29.50 Ma is defined by the LO of *Lychnocanium* aff. *conicum.* The LO of *Clinorhabdus robusta* sets the base of aRP17 and is found at 28.20 Ma. The last zone within the Oligocene is aRP18, it has a base age of 26.10–19.0 Ma, and the base of it is defined by the HO of *Axoprunum? irregularis*.

## 2.4 Dinoflagellate cyst zones

Global Oligocene dinoflagellate cysts zones cannot be defined easily as species ranges strongly vary as a function of environmental gradients and latitude (Bijl, 2022). Stratigraphic ranges of many dinocyst species and zones are therefore typically regionally constrained although several globally recorded events do occur across the Oligocene (e.g., Bijl et al., 2018; Van Simaeys et al., 2005; Wilpshaar et al., 1996). Still, the world-wide significance of Oligocene dinocyst biostratigraphic events requires more detailed taxonomic and stratigraphic analyses (Bijl 2022). Interestingly, the EOT and OMT represent strong dinocyst extinction phases and can thus typically be easily identified in sedimentary records (e.g.,Brinkhuis et al., 1992; Brinkhuis & Biffi, 1993). Contrary to the calcareous microfossil events used to define stage boundaries, the recorded dinocyst events correlate tightly to the strong climate perturbations in these transitions (e.g., Houben et al., 2011, 2013; van Mourik & Brinkhuis, 2005).



The Oligocene zones listed in the GTS2020 are focused around the dinocyst assemblages of the mid-latitudes from the NH notably from the North Sea (e.g., King, 2016; Mudge & Bujak, 1996; Powell et al., 1996). The first zonation for the Oligocene (D12–D16) in the mid-latitudes of the NH stems from the GTS2004 (Gradstein et al., 2004) which was done by Powell and

Brinkhuis (2004). This zonation was later revised by King (2016) and split up into 9 sub-zonations (DE20b, DO1–DO7, DM1a) for the Oligocene. The top of the latest Eocene zone DE20b is defined by the LO of *Areosphaeridium diktyoplokum* at 33.757 Ma, corresponding closely to global climate deterioration. DO1 is split into three subzones (a,b,c), of which the base of DO1b is defined by the FO of *Wetzeliella gochtii* top and the base of DO1b correlates to the LO of *Glaphyrocysta semitecta*. The base of DO2 is defined by the FO of *Chiropteridium spp.* and the zone DO2 ends with the LO of *Phthanoperidinium comatum.*

DO2 is also divided into three subzones with the top and bottom of DO2b being defined by the FO and LO of *Spiniferites manumii*. The top of DO3 and base of DO4 concurs with the LCO of *Enneadocysta pectiniformis* as well as the FO of *Apteodinium spiridoides*. DO5 also comes with three subzones, with the top of DO5a defined by the FO of *Distatodinium biffii*, the top of DO5b by the FO of *Svalbardella cooksoniae*, the top of DO5c by the LO of *Svalbardella cooksoniae* and the DO5 zones base by the LO of *Rhombodinium draco*. DO6 is a short zone with its base at the LO of *Enneadocysta pectiniformis.*

The zone DO7 is divided into two subsections with the base of DO7a correlating to the FO of *Triphragmadinium demaniae* and the base of DO7b to the FO of *Tuberculodinium vancampoae.* The last subzone within the Oligocene is DM1a which ends at the OMB with the FO of *Invertocysta tabulata.*

Two environmentally driven dinocyst migration events have also been documented, notably involving the boreal Atlantic dinoflagellate *Svalbardella* interpreted as a cold-water indicator (Head and Norris, 1989; Sluijs et al., 2005). Its conspicuous

southward migration at Oi-2, even to the SH has thus been interpreted as an indication of cold phases (Head and Norris, 1989; Van Simaeys et al., 2005). The first of the two intervals correlate with the EOIS cooling at the end of Chron C13n (Śliwińska and Heilmann-Clausen, 2011). The second *Svalbardella* interval is associated with the Oi-2b cooling event (Śliwińska et al., 2010; Van Simaeys et al., 2005).

## 3. Boundary Conditions for Oligocene climate

### 3.1 Geographical Boundary Conditions

Oligocene plate tectonic boundary conditions differed from the modern configuration regarding several regions that are relevant to climate (Fig. 3). Specifically, plate tectonic movements may have been important for oceanographical change, influencing regional climate through changes in meridional and zonal heat transport. We here discuss the most prominent tectonic changes.



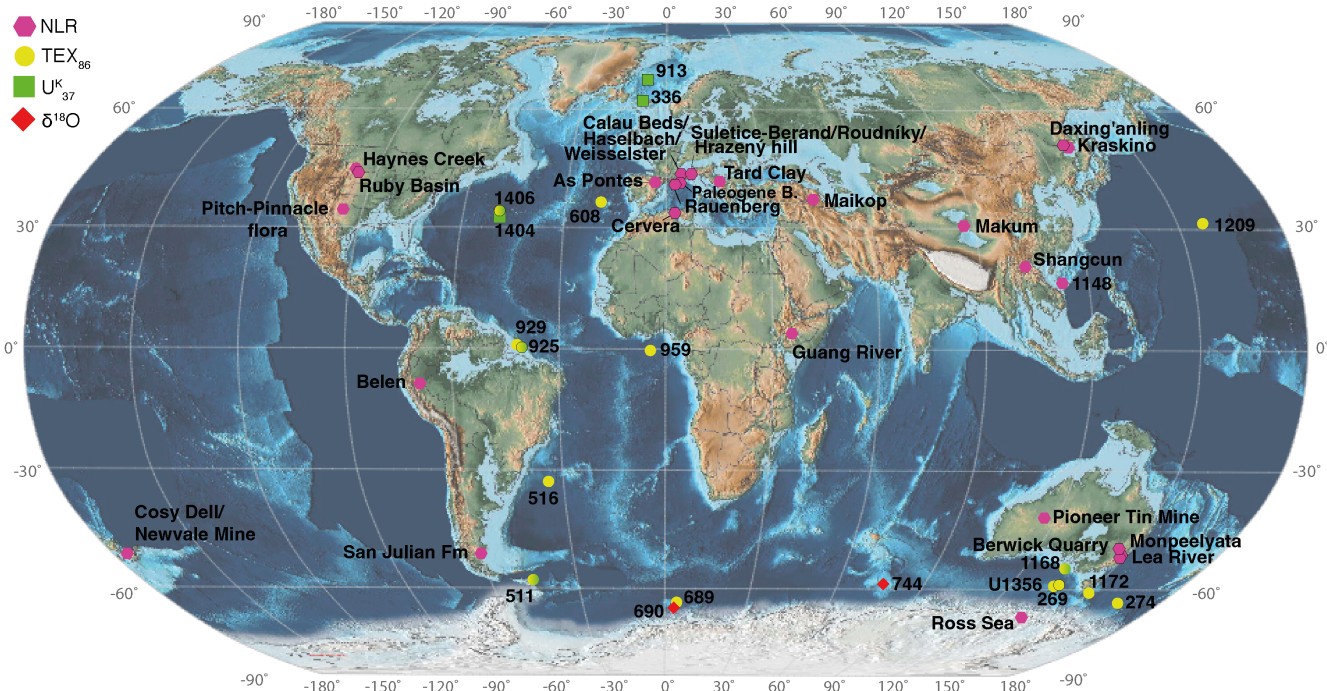


**Figure 3: Paleogeographic reconstruction of the Oligocene (~28 Ma). Yellow dots: Tex₈₆ based data; purple hexagons: Nearest Living Relative (NRL) data; green squares: U$^K_{37}$ data; red diamonds: δ¹⁸O data. Map created using Gplates, using the** Scotese & Wright (2018) **plate rotation.**

One of the most discussed tectonic changes since the Oligocene is the uplift of the Tibetan Region (consisting of the Tibetan

Plateau, the Himalaya and the Hengduan Mountains). Although the collision between India and the Eurasian plate predates the

Oligocene (60–50 Ma; van Hinsbergen, 2022; Wang et al., 2014), continued collision created further uplift of the Himalaya

and the Tibetan Plateau also in the Oligocene. The climatic consequences of the uplift are investigated both regionally (e.g.,

SE Asia; Ding et al., 2017; Su et al., 2019) and globally as a source of chemical weathering during the Cenozoic (e.g., Raymo

& Ruddiman, 1992). Today the Tibetan Region has an average elevation of over 4.5km. During the Oligocene the elevation

of the Tibetan Region was between 2.3 and 3km, with central Tibet most likely at similar altitudes as today (Spicer et al., 2020,

2021b, a; Su et al., 2019).

Another important tectonic event during the Oligocene was the convergence of the European and Adriatic plate which led to

the formation of the Alpine system. While the collisional stage of the Alps began in the earliest Paleocene (65 Ma), during the

Oligocene a slab from the subducted oceanic European lithosphere broke off, which resulted in the rapid and continued uplift

of the Alps, which lasted until today but became stable in the mid Miocene (Meschede and Warr, 2019). This resulted in an

Oligocene uplift of ~1000m of the Alpine area, from an average elevation of  <1km to 2km (Dielforder, 2017; Winterberg et

al., 2020). Due to the strong tectonic changes around the eastern Alps and the Tibetan Region during the Oligocene, the so

called 'Paratethys' (Laskarev, 1924), which reached all the way from the western Molasse Basin in Switzerland to the now



mostly dried up lake Aral between Kazakhstan and Uzbekistan, became a semi-isolated inland sea at the beginning of the
Oligocene (Schulz et al., 2005; Steininger and Wessely, 1999). The Paratethys consisted of a series of adjacent sedimentary basins, of which the interconnections and connections were rather unstable resulting in the separation of the Paratethys into three main parts: the Western (Alpine), Central (Balkan) and Eastern (Caucasian) Paratethys (Kováč, 2017; Palcu and Krijgsman, 2023; Rögl, 1998). A connection to the Mediterranean might have been established in the late Oligocene, connecting the Paratethys to the main oceans (Kováč, 2017).

Oceanic gateways in the Southern Ocean, including the Drake Passage (DP) and the Tasmanian Gateway, underwent tectonic changes in the Paleogene that strongly affected regional ocean circulation and associated heat and salt transport and biogeography, as inferred from sedimentary data (e.g., Kennett, 1977; Murphy & Kennett, 1986). However, the extent to which the opening of these gateways affected Cenozoic cooling, either globally or regionally, remains the subject of debate (Houben et al., 2019; Huber et al., 2004; Kennett, 1977; Sauermilch et al., 2021; Scher and Martin, 2006; Toumoulin et al., 2020). While
exact age estimates for the first opening of the DP lay most likely in the middle to late Eocene (∼50 Ma), it remains unclear whether the DP opened once or experienced intermittent closures that resulted in staggered throughflow over several tens of millions of years. The DP opened due to tectonic processes between the South American, the Antarctic and the Central Scotia Plate (Eagles, 2016), and involved a complex opening of isolated ocean basins, which became oceanographically connected into one deep throughflow during the Oligocene (~26 Ma; van de Lagemaat et al., 2021). During the early to middle Oligocene
(34–26 Ma), subduction initiated between South America and the Scotia Plate (Crameri et al., 2020), which led to the opening and deepening of several ocean basins in the area between 29.5 and 21.2 Ma (van de Lagemaat et al., 2021). This tectonic development facilitated the formation of a deeper DP gateway (Maldonado et al., 2014).

Although spreading between Australia and Antarctica initiated in the Cretaceous, the South Tasman Rise connected the continents until the latest Eocene (see overview in Bijl et al., 2021). Dinoflagellate cyst biogeographical evidence suggests
shallow water connections between the Australo-Antarctic Gulf and the southwest Pacific initiated close to the early–middle Eocene transition (Bijl et al., 2013). Lithological evidence for rapid deepening of the South Tasman Rise at ~35.7 Ma (Stickley et al., 2004) was later found to be a Southern Ocean-wide phenomenon which initiated the throughflow of both a proto-Antarctic Circumpolar Current but also of a vigorous Antarctic Counter Current (Houben et al., 2019). It is generally accepted that the Tasmanian Gateway was open to deep waters by Oligocene times (Stickley et al., 2004) but the Australian continent
obstructed the optimal flow of strong circumpolar ocean currents until the late Neogene (Evangelinos et al., 2022; Hill et al., 2013; Sauermilch et al., 2021).

## 3.2 Ocean Circulation

Using marine magnetic data, Barker & Burrell (1977) found that the opening of the Southern Ocean gateways (SOG) (i.e., Drake Passage and Tasmanian gateway) preconditioned the formation of the Antarctic Circumpolar Current (ACC) between
Antarctica, South America, and Australia. As Earth's strongest ocean current, the modern ACC is not only responsible for the regulation of heat and carbon exchange from and to the atmosphere, but it also influences deep-water formation and nutrient



distribution (Cox, 1989; Scher et al., 2015). The ACC encircles Antarctica and in doing so connects the deep waters of the Pacific, Indian and Atlantic Ocean. Models using middle Eocene to early Oligocene geographies and $CO_2$ values have sought to understand the influence of the SOG openings on the global ocean (e.g., Goldner et al., 2014; Hutchinson et al., 2018;
Kennedy et al., 2015; Kennedy-Asser et al., 2019; Sauermilch et al., 2021). Model simulations suggest that as soon as the DP opened, a weak current (the proto-ACC) from the Pacific to the Atlantic would have established (Ladant et al., 2014). Additionally, the coupled model of Toggweiler & Bjornsson (2000) shows that winds around Antarctica raise cold dense water, cooling the region. This upwelled water then becomes fresher and warmer as it moves northwards due to Ekman transport. North of the ACC this lighter and warm water is transported downwards into the thermocline. This thickens the lower
thermocline and creates a bigger density contrast across the Icelandic sills, ultimately enhancing the formation of the North Atlantic Deep Water (NADW). Subsequently, the NADW cools down the water, facilitating the southward transport of the water. The model of Toggweiler & Bjornsson (2000) shows that the DP opening thus might have led to a cooling of the air and oceans around Antarctica of around 3°C. The water in the SH takes up the solar heat, which is then transported northwards where it is released, consequently warming up the NH by the same amount the SH was cooled. (Toggweiler and Bjornsson,
305  2000).

Lagabrielle et al. (2009) also discuss the influence of the proto-ACC on the formation/strength of the Northern Component Water (NCW, later turns into the North Atlantic Deep Water), which brings water from the NH to the Southern Ocean. Exactly when the formation of NCW began is unclear, but around 34 Ma the north Atlantic deepened rapidly due to its separation from Greenland in response to the Iceland mantle plume collapse, a significant deep-water production started in the NH (Lagabrielle
et al., 2009; Via and Thomas, 2006). Hence, along with tectonic changes in the North Atlantic region, the proto-ACC contributed to the inception of NCW/ North Atlantic Deep Water (NADW) and modulated its strength.

In the Eocene the SOGs were not open to deep ocean circulation. Rather, shallow ocean connections south of 60 °S allowed for a westward Antarctic Counter Current (e.g., Bijl et al., 2013; Houben et al., 2019). Although SOGs progressively opened in the Oligocene (Stickley et al., 2004), oceanographic changes associated with that were restricted to Southern Ocean (Houben
et al., 2019), and there was little effect on the Southern Ocean oceanography for the remainder of the Oligocene (Evangelinos et al., 2020; Hill et al., 2013; Hoem et al., 2021; Wright et al., 2018). Only in the late Oligocene, Southern Ocean latitudinal SST gradients increased and perhaps the ACC strengthened, due to deep opening of Drake Passage (Hoem et al., 2022), although the ACC weakened again during the Miocene Climatic Optimum (Evangelinos et al., 2022; Sangiorgi et al., 2018).

## 3.4 Carbon Cycle

Cooling and ice sheet growth during the EOT was accompanied by a ~1‰ rise in deep ocean benthic foraminifer $\delta^{13}C$ from around 0.5 to 1.5‰ (Fig. 1c), which peaked during the EOIS (Coxall and Wilson, 2011). This increase in $\delta^{13}C$ lags ~20–30 ka behind the recorded EOIS $\delta^{18}O$ increase at ~33.71 Ma (Coxall and Wilson, 2011). After the EOIS the $\delta^{13}C$ decreases towards the middle Oligocene to ~0.2‰, only to increase again after to ~1‰ at the OMT. Several mechanisms have been invoked to explain these trends including changes in silicate or shelf carbonate weathering (Zachos and Kump, 2005), carbonate and





organic carbon burial in the deep ocean (e.g., Merico et al., 2008), expansion of carbon capacitors (e.g., Armstrong McKay et al., 2016) as well as an increase in ocean mixing (e.g. (Miller et al., 2009). In order to understand the Oligocene carbon cycle, data on relative changes in silicate weathering, carbon burial on land and in ocean sediments and the carbonate compensation depth are needed (e.g., Berner et al., 1983).

An increase in ocean mixing would have led to increased deep oceans ventilation, ultimately resulting in reduced deep-ocean

acidity and thus a deepened Calcite Compensation Depth (CCD) (Miller et al., 2009). Additionally, it would have led to an increased plankton productivity in the Southern Ocean due to the upwelling of nutrients (Salamy and Zachos, 1999; Scher and Martin, 2006; Zachos and Kump, 2005). Salamy & Zachos, (1999) found a profound local (Southern Indian Ocean) increase in mass accumulation rates of biogenic opal at the cost of biogenic carbonate across the Oi-1a. They interpreted this as an increase in primary production, likely seasonal, due to upwelling in the Southern Ocean following Antarctic glaciation. An

increase in seasonal production along the Antarctic margin may have been associated with an increased flux of organic carbon to the ocean sediment, increasing organic carbon burial, and explaining the recorded $\delta^{13}C$ peak around the EOIS (Zachos and Kump, 2005). Also if productivity was shifted to upwelling cells, organic carbon preservation would have increased in the deep oceans (Zachos et al., 1996). Another explanation for the rise in oceanic $\delta^{13}C$ at the EOIS is an increase in weathering of continental silicates and thus increased $Ca^{2+}$ input into the oceans (Zachos and Kump, 2005). Increased $Ca^{2+}$ and $HCO^-$ would

lead to higher ocean alkalinity, which would lead to a deepening of the CCD (Farkaš et al., 2007; Komar and Zeebe, 2011). Furthermore, the growth of continental ice on Antarctica caused the global average sea level to drop considerably (Houben et al., 2012; Miller et al., 2008), with a regional rise near the ice sheet (Stocchi et al., 2013). Lower sea levels not only would expose more carbonate rich rocks to weathering but also reduces the area for marine carbon deposition on continental shelves and leads to increased carbonate deposition in deep oceanic basins (Armstrong McKay et al., 2016; Tripati et al., 2005). With

temporarily increased $^{13}C$ enriched carbonate erosion on shelves and a continuous alkalinity supply by rivers due to silicate weathering on land, bypassing the shelves, calcite saturation increases leading to a drop in the CCD until deep ocean burial compensates for the reduced shelf burial (Armstrong McKay et al., 2016; Merico et al., 2008; Wade et al., 2020). Lastly, more carbon reservoirs such as permafrost, ocean methane hydrates, peat and wetlands develop due to the colder temperatures and ice expansion. These capacitors store more carbon leading to an atmospheric $CO_2$ drawdown and thus a positive $\delta^{13}C$

(Armstrong McKay et al., 2016). Collectively, it is most likely that the $\delta^{13}C$ increase during the EOT was caused by a combination of increased shelf to basin carbonate burial, carbonate weathering, as well as increased ocean ventilation and storage of $^{12}C$ in carbon reservoirs.

Only a few records of atmospheric $p\mathrm{CO}_2$ cover the Oligocene entirely, most are focused on the EOT and the OMT, or, are of low resolution. Trends are quite inconsistent between records and proxies. The available records for the Oligocene are based

on higher plant leaf gas exchange, phytoplankton $^{13}C$-fractionation, and boron isotopes (Fig. 1). Pagani et al. (2005) were the first to produce a $p\mathrm{CO}_2$ record for the Oligocene using stable carbon isotopic fractionation of di-unsaturated alkenones extracted from various Deep Sea Drilling Program (DSDP) and Ocean Drilling Program (ODP) sediments. They recorded decreasing $\mathrm{CO}_2$ throughout the Oligocene from ~1500ppm at the EOT to modern levels by the late Oligocene. Pagani et al.



(2011) evaluated regional differences in $CO_2$ and $CO_2$ trends over the EOT by contrasting alkenone carbon isotope values from
six DSDP and ODP Sites. The estimated $CO_2$ values yielded highly variable results among the different sites showing a general
atmospheric $pCO_2$ decline from around 1200 to around 600 ppm throughout the Oligocene, with $pCO_2$ decreasing around 40%
at the EOT. Zhang et al. (2013) critically evaluated confounding factors of the alkenone $pCO_2$ proxy and excluded data from
several locations, arriving at a continuous $CO_2$ record covering the past 40 Ma based on di-unsaturated alkenone $^{13}$C-
fractionation at ODP Site 925. The general findings of Zhang et al. (2013) agreed with the $pCO_2$ trends reported in Pagani et
al. (2005, 2011) but showed $pCO_2$ values to decrease from ~1000 ppm at the EOT to ~400 ppm in the late Oligocene. Lastly,
Witkowski et al. (2018) compiled the longest consecutive $pCO_2$ record of the past ~100 Ma, solely based on phytane $^{13}$C-
fractionation from marine sediment and oil samples. Their results concur with the findings of Zhang et al. (2013), showing
$pCO_2$ ranges from ~600–1000 ppm throughout the Oligocene with a decreasing trend from the EOT towards the OMT (Fig.
1). It should be noted that absolute values derived from alkenone based proxies may show a muted signal, thus potentially
inaccurately reflecting atmospheric $CO_2$ values (Badger et al., 2019).

Both Roth-Nebelsick et al. (2014) and Moraweck et al. (2019) used fossil leaf stomata to reconstruct Oligocene atmospheric
$CO_2$ levels. Oligocene fossil leaves of *Platanus neptuni* from various sites in Saxony (Germany) suggest lower $pCO_2$ levels
than the alkenone-based results with a modelled range of ~400–600 ppm for the Oligocene (Roth-Nebelsick et al., 2014).
Moraweck et al. (2019) reconstructed $pCO_2$ from the middle Eocene to the Oligocene using *P. neptuni* and *Rhodomyrtophyllum*
*reticulosum* leaves from 7 central European sites. They found a similar $pCO_2$ range as Roth-Nebelsick et al. (2014) with values
also varying between ~400–600 ppm in the Oligocene. Greenop et al. (2019) created the only available boron isotope-based
$CO_2$ record, however they only 16esults on the OMT. While Greenop et al. (2019) did not find a strong decreasing trend over
the OMT, they generally found low, stable values ranging from around 220 to 350ppm, which then increased to around 400
ppm after the OMT.
Despite some variability in all these Oligocene records, they consistently show a significant $pCO_2$ drop at the EOT and suggest
that the $pCO_2$ likely decreased during the Oligocene towards the Miocene.

## 4. Climate Proxy data

We compiled marine and terrestrial climate proxy data to assess long-term trends and variability in climate across the
Oligocene. For sea surface temperatures, we have added recently published records to the compilation of O'Brien et al. (2020).
To assess terrestrial climate, we compiled published records of fossil plant remains, notably pollen, spores, and macro-remains
(appendix Table A1). Where the fossil plant remains had been assigned taxonomic affinities, the nearest living relatives (NLR)
were determined and used as input for NLR-based probability density modeling (Huurdeman et al., 2021; Willard et al., 2019),
to assess terrestrial paleoclimate. We adopt the age determination from the original sources (appendix Table A1), corrected
for the GTS 2020 stage boundaries (Gradstein, 2020), where absolute age determination is unavailable an average age was
taken. Based on the distribution of the NLR for each fossil species, the probability of plant co-existence in an assemblage is



calculated for 60,000 combinations of mean annual temperature (MAT), winter mean temperatures (WinT), mean annual precipitation (MAP) and driest month precipitation (DMP). Up to 20 different plant taxa were compared at a time and where there were more than 20 taxa, sets of 10 were randomly chosen to maximize data variability. We report the highest probability climate combination and the uncertainty range is based on those climatic combinations with a probability of ≥2.5% the
maximum probability combination.

We found 28 vegetation reconstructions of sufficient quality to assess paleoclimate using the NLR method (see appendix Table A1). The results can be assigned several potential quality "flags" based on diversity, depositional environment, and taxonomical assignments. First, low convergence of multiple simulations of the same flora may suggest that the climate niche of one or multiple taxa has changed since the Oligocene (Reichgelt et al., 2023). Additionally, some floras had fewer than 20
taxa recorded, for which convergence could not be tested. Second, microfloras (pollen and spores) likely include upland or even extra-basinal input and are therefore less indicative of local climatic conditions than macrofloras (leaves, fruits, flowers) (Reichgelt et al., 2023). Third and finally, some paleobotanical studies assign fossils to parataxa based on limited anatomical evidence, or using literature that is inappropriate for the study region. The majority of the data derives from NH mid-latitudes, a handful from SH mid-latitudes and two datasets from the tropical realm. The absence of high latitude data may be partly due
to the lack of vegetation due to cool conditions. However, there are pollen assemblages in sediments from the Antarctic margin (e.g., Askin & Raine, 2000; Prebble et al., 2005; Raine & Askin, 2001), but to our knowledge no quantitative data suitable for our NLR method has been generated for any high-latitude site.

## 4.1 Temperature

### 4.1.1 Continental Mean Annual Temperature (MAT)

The data produced by the NLR allows for a first assessment of the general Oligocene meridional temperature gradient. Unfortunately, the data are too sparse to assess trends or variability at any location, on all time scales (Fig. 4). Although the sparsity of data from low and mid-latitude datasets limits our view on global gradients, the mid-latitude data can be compared to the modern and model simulations of the Oligocene, as well as to reconstructed Eocene and Miocene gradients.

The two low latitude data points suggest MATs of around 25°C (±1.5 °C). Except for one data point in the NH that indicates
a temperature of ~24 °C, all other mid-latitude MAT reconstructions in both the Northern and SH are mostly between 12 and 17 °C, with an average error of ±2.9°C (Fig. 4). At first sight, Oligocene low latitude MATs are, on average, similar to modern MAT for the same latitudes, but the NLR method is based on modern distributions and therefore reconstructed temperatures cannot exceed the global temperature maximum. Paleobotanical temperature reconstructions from the tropics are susceptible to this problem, and should therefore be considered minimum estimates. Mid-latitude MAT, particularly in the SH, during the
Oligocene are generally higher (up to 16°C) than modern temperatures.

The reconstructed winter temperatures (WinT) range from ~23°C in the lower latitudes to ~3°C in the highest latitude samples (~52°N). Temperatures around the equator reveal limited change in winter cooling (~1.8°C), while at higher latitudes the



difference in temperature between WinT and MAT can be up to nearly 8°C. Compared to modern values, Oligocene WinT show the same trend as MAT, with similar values (possibly underestimations) around the equator and warmer temperatures

around the mid-latitudes (Fig. 4). The seasonality trends in modern times are similar to the Oligocene, with WinT of the higher latitudes dropping by up to ~12°C compared to MAT. Just as in the Oligocene, modern WinT have a larger range at higher latitudes.

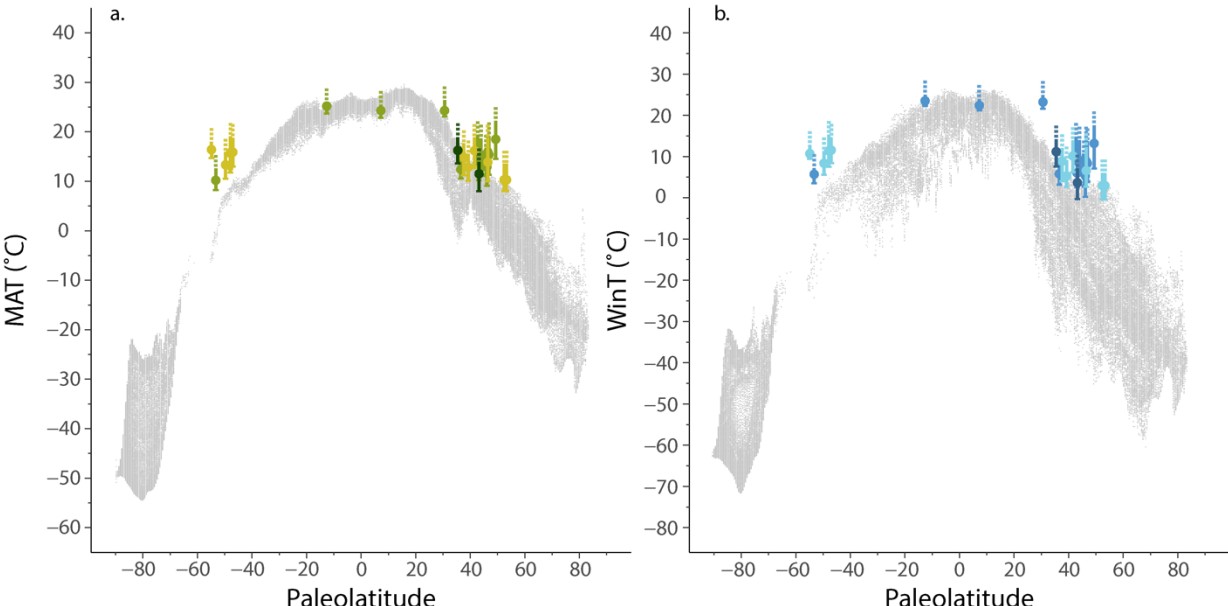

**Figure 4: a. Mean annual temperature (MAT) plot over paleolatitudes, in grey: Pre-industrial (1900) MAT from Matsuura & Willmott, (2018); b. Winter Temperature (WinT) plot over paleolatitudes, in brown: Pre-industrial (1900) WinT from Matsuura & Willmott, (2018). Darker colours represent a higher analytical certainty of the used site, data with low reliability were excluded (see appendix Table A1).**

### 4.1.2 Sea Surface Temperatures (SSTs)

Sea Surface Temperatures (SSTs) from three different proxies ($U^{k'}_{37}$, $TEX_{86}$ and biogenic calcite $\delta^{18}O$) were compiled for the low-, mid-, and high latitudes of the Oligocene (Fig. 5). The alkenone-based SST reconstruction ($U^{k'}_{37}$), relies on the temperature dependence of di- and tri-$C_{37}$ ketones (Prahl and Wakeham, 1987). At $U^{k'}_{37}$ values of ~0.9 (at SSTs >27 °C) the proportion of the tri-unsaturated $C_{37}$ alkenone becomes very low — virtually absent and/or undetectable, setting an upper limit for the application of this proxy (Tierney and Tingley, 2018). In addition, the low proportion of this alkenone introduces

analytical uncertainties that cause noise. Consequently, we consider all $U^{k'}_{37}$ values >0.9 to represent SSTs at or above 27 °C. Following the recommendations by Hollis et al. (2019), we use the calibration of Müller et al. (1998, see appendix Table A2).





The TEX$_{86}$ palaeothermometer is based on the temperature-sensitivity of marine thaumarchaeotal membrane lipid (isoprenoidal glycerol dialkyl glycerol tetraether (isoGDGT) distributions (Schouten et al., 2002). The proportion of GDGTs containing a greater number of cyclopentane rings increases with higher temperatures, and can thus be used to calculate SSTs
using a modern surface sediment calibration (Wuchter et al., 2004). Discussion remains on how TEX$_{86}$ should be calibrated to represent seawater temperatures. The surface sediment calibration dataset shows virtually no response to temperature below 15 °C and it is debated if the response at the high-temperature end of the modern ocean — analogous to warmer climates in the past — can be assumed to be linear (e.g.,O'Brien et al., 2017; Tierney & Tingley, 2014) or decreases exponentially (e.g., Cramwinckel et al., 2018; Jones and Jones, 2020). Moreover, isoGDGTs are barely produced in the mixed layer — they peak
at ~50–200 m depth, and sometimes somewhat deeper (e.g., Hurley et al., 2018; van der Weijst et al., 2022). Most calibrations include surface ocean temperatures in their calibration dataset, leading to an overestimation of the proxy slope (Ho and Laepple, 2016). As the Oligocene is most likely warmer than today, we therefore prefer a conservative approach to assess SST, using an exponential calibration that has a drop in proxy-response at higher temperature levels. Even though it is associated with significant statistical problems (Tierney and Tingley, 2014), we use the TEX$_{86}^{H}$ of Kim et al. (2010, see appendix Table
A2) to assess SST rather than a linear model. Linear models produce much higher SSTs in the Oligocene TEX$_{86}$ range (Hollis et al., 2019). Moreover, any SST calibration assumes a similar relationship between surface temperature and the isoGDGT export zone in both modern and ancient oceans. Given the above uncertainties, it should be noted that absolute SSTs come with large uncertainties and should be taken with care.

Foraminifer oxygen isotope ratios were also used to estimate Oligocene SSTs. This method is based on the direct correlation
between the temperature dependent fractionation of the oxygen isotopes $^{16}$O and $^{18}$O) into biogenic calcite of foraminifera (Shackleton, 1974). Here, the calibration of Kim & O'Neil (1997) is used because it is based on inorganic calcite precipitated at temperatures between 10 and 40 °C and produces reliable results for foraminifera (Hollis et al., 2019). Using this calibration, an over estimation of up to 1.5 °C has to be taken into consideration, due to algal photosymbionts which modify the pH of the calcifying microenvironment (Spero and Williams, 1988).

Most available SST data are from the mid-paleolatitudes; records for the low- and high latitudes, especially of the NH, are scarce. Moreover, most records have low temporal resolution or cover only specific segments of the Oligocene (namely the EOT and OMT). The high latitude SSTs vary from 9.8 to 25.1 °C. It is worth noting that these records are restricted to latitudes between 67°N and 68°S. SST estimates from mid latitude locations have the largest temperature range, 6.0–32.1 °C, while SST estimates from low latitude sites span a narrower temperature range, 23.7–34.4 °C. The mid latitude SSTs show a slight
increase (1–2 °C) between 34 Ma and ~27 Ma, followed by a small decrease of 1–2 °C towards 23 Ma (Fig. 5). However, overall, there is a remarkable absence of long-term trends in these SST records.



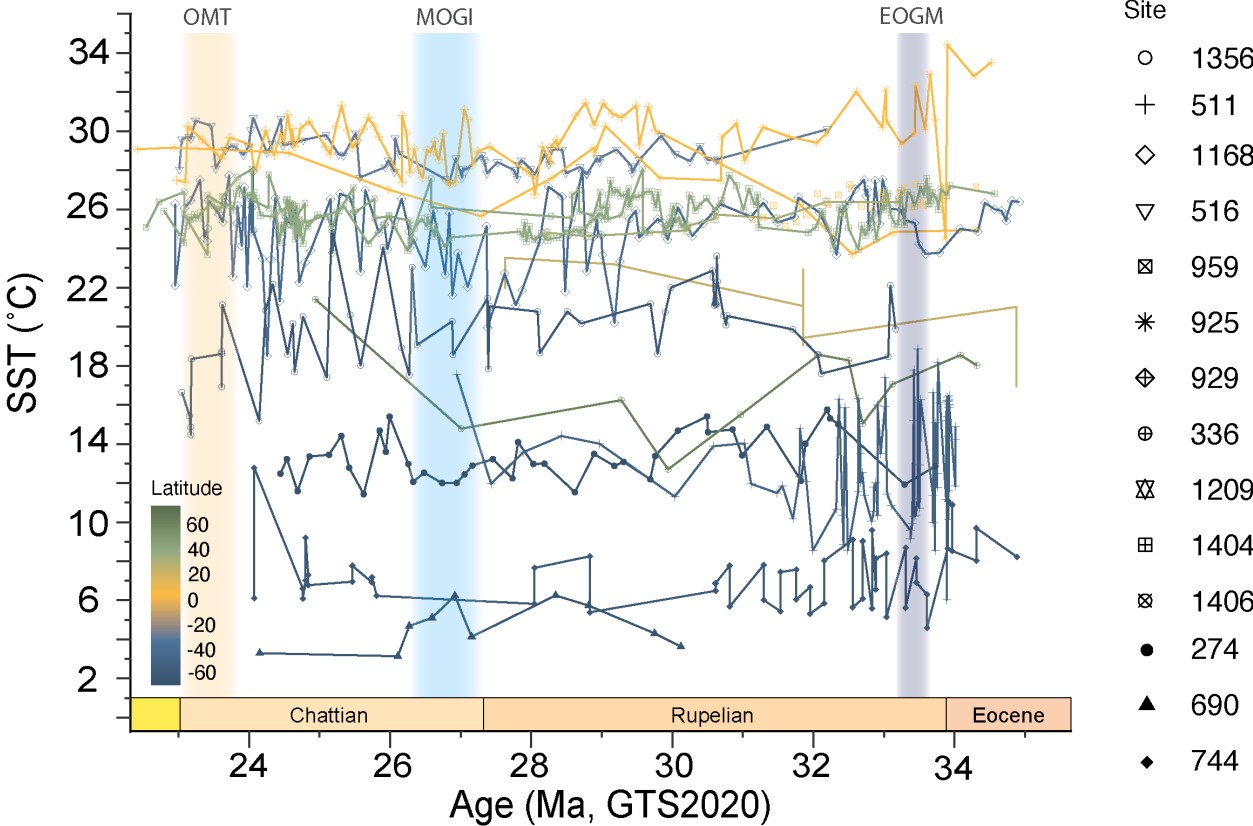

**Figure 5: SST compilation for the Oligocene via O'Brien et al. (2020) and references therein. A linear interpolation was used between**
**datapoints. Blue sites: SH high latitude sites; yellow sites: low latitude sites; green sites: NH high latitude sites. See Fig. 3 for exact**
**site locations and appendix Table A3 for references.**

To assess long-term changes in temperature and meridional temperature gradients, we analyze data from three time slices:

33.7–33.2 Ma, 27.3–26.3 Ma, and 23.9–23 Ma, corresponding to the EOGM, MOGI, and OMT, which are averaged for 33.4

Ma, 26.8 Ma, and 23.4 Ma, respectively. When SSTs are corrected for paleolatitude (see appendix Table A3) Oligocene SSTs

are closer to late Eocene (~38 Ma) than to modern values (Fig. 6, brown shaded area). This is especially apparent in the

Southern Ocean where Oligocene SSTs are up to 10 °C warmer than modern. The high latitudes of the NH are challenging to

reconstruct due to data scarcity. However, the data available indicate that Oligocene SSTs were ~2 °C colder than Eocene

SSTs but still ~4 °C warmer than current records. In contrast, low latitude SST reconstructions show minimal differences,

yielding similar temperature estimates for both the Oligocene and the Eocene.






**Figure 6: Sea surface temperatures (SSTs) over Paleolatitudes for 33.4 Ma (EOGM, black dots), 26.8 Ma (MOGI, blue triangles), 23.4 Ma (OMT, orange squares). Brown shaded area:** Baatsen et al., (2020) **SST record for the late Eocene (38 Ma). Grey area: Pre-industrial (1900) SST over latitude (Huang et al., 2015). Thick vertical error bars show the SST standard deviation, thin vertical error bars represent the calibration error for each proxy. Larger symbols represent a higher data resolution, with larger symbols**

**representing more data used and smaller points where less data was available. See appendix Table A3 for data referral and references used.**



## 4.2 Precipitation

Mean annual precipitation (MAP) and driest month precipitation (DMP) were derived using the NLR approach (Table 1). The reconstructed MAP shows a range from ~850 mm/yr to 1750 mm/yr. The SH mid-latitudes show a generally higher MAP

(~1200–1750 mm/yr) for the Oligocene compared to the NH (~850–1650 mm/yr). This differs from modern MAP values, where there is not a big discrepancy between SH and NH MAP. Generally, the Oligocene MAP values are similar to the modern values. The few tropical datapoints might suggest relatively dry conditions but are not outside the range of modern values. The values for the driest month range from ~10 mm/yr to 85 mm/yr with generally lower values around the equator and the NH (~10–45 mm/yr) and higher DMP on the SH (~20–85 mm/yr). The DMP values around the equator are generally

lower compared to modern values. Similarly, as in the Oligocene, modern SH DMP are, on average, higher than NH DMP values.

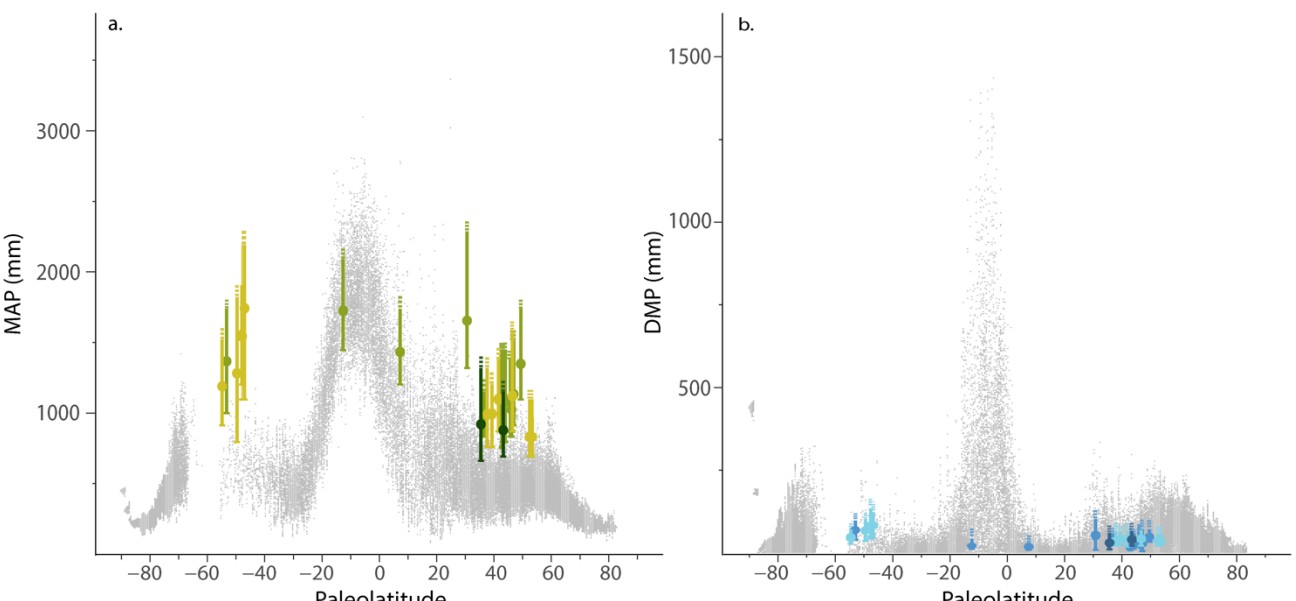

**Figure 7: a. Mean annual precipitation (MAP) plot over paleolatitudes, in grey: Pre-industrial (1900) MAP via Matsuura & Willmott, (2018); b. Driest month precipitation (DMP) plot over paleolatitudes, in grey: Pre-industrial (1900) DMP via Matsuura & Willmott, (2018). Darker colours represent a higher analytical certainty of the used site, data with low reliability was excluded (see appendix Table A1).**


## 4.3 Data–Model Comparisons

The compiled surface temperature and precipitation data was regionally compared against the results from paleoclimate model simulations (Figs. 8 and 9). Similar to O'Brien et al. (2020), two sets of modelling experiments were used: one from the NCAR

Community Earth System Model version 1.0 (CESM1.0) and the other from the UK Hadley Centre Coupled Model version 3 (HadCM3L). We compare data and simulations for three time slices (a. Early Oligocene, 33.9–33.0 Ma; b. Middle Oligocene,



33.0–26.5 Ma; c. Late Oligocene, 25.0–23.5 Ma, Fig. 8). For each time slice, the model ensemble mean is used to compare with the data, and the modelling details of the ensemble members are seen in Methods and Supporting Information Table S1 of O'Brien et al. (2020).

### 4.3.1 Temperature proxy to model comparison

The comparison of sea and land temperatures of all three time slices, show that the mid and high latitude proxy data generally suggests warmer local conditions than simulations predict (Fig. 8). For all investigated time slices, that discrepancy is largest in the North Atlantic and southwest Pacific. Additionally, for every time slice, the tropics seem to be warmer in model simulations than what actual local proxy data find, with the most extreme discrepancies in southeastern Asia. For the early and middle Oligocene, there is a visible difference between the modelled higher latitudes and measured data. In both the early and middle Oligocene, the higher latitudes seem to be a lot warmer (up to 20°C) in the proxy data than what the models predict. The lower latitudes on the other hand for both the early and middle Oligocene seem somewhat colder (around 5°C) than what models predict. In the early Oligocene, North America generally shows a similar temperature range as the European sites, with proxy data indicating warmer conditions compared to the model. This shifts in the middle Oligocene, where most of the recorded North American sites are colder than what the models predict. The late Oligocene seems to have a similar offset towards the SH high latitudes, with reconstructed temperatures being up to nearly 20°C warmer than the model results. Due to the lack of proxy data in the NH high latitudes and the tropics, temperatures differences between models and records cannot be determined for the late Oligocene.





**Figure 8: Oligocene temperature data to model comparisons. All proxy data was compared to ensemble means from HadCM3L (UK) and CESM models. (A–I) Sea surface and Land temperature data-model comparisons for three Oligocene time slices. The data is both displayed in a spatial (A, B, D, E, G, H) and zonal mean (C, F, I) of the different temperatures. Differences in A, B, D, E, G, H are calculated as a pointwise difference between the proxy mean value and the model annual mean and then plotted as points of differences. The pink and blue ribbons in C, F and I represent the maximum and minimum differences associated with the zonal means. All proxy data are shown in appendix Table A4.**



### 4.3.2 Precipitation proxy to model comparison

The precipitation comparison of proxy data to modelled simulations show that for all three simulated times models mostly slightly underestimate the daily precipitation on a global scale (Fig. 9). In particular, the SH mid to lower latitudes seem to be much drier in the models that what the proxy data suggests. For the early Oligocene only NH tropical and NH mid latitude data are available, indicating a slightly wetter climate around the European continent. Whereas eastern Asia and western North America appear to be drier than model predictions. Due to limited proxy data, we cannot make definitive statements about the early Oligocene in North America and eastern Asia. In the middle Oligocene, although more proxy data is available compared to the early Oligocene, the trends are similar. Compared to the model results, the proxy data suggest wetter climates in Europe and generally drier conditions in North America and eastern Asia, with the exception of one eastern Asian site (Makum Coal Field) which has a wetter climate compared to the model, yet that might be a localized effect. Additionally, in the middle Oligocene, model simulations appear to underestimate precipitation in SH mid- to high latitudes (Fig. 9). This underestimation is also observed in the late Oligocene. Similar to the early and mid-Oligocene, late Oligocene precipitation over central Europe and the region corresponding to today's Middle East also appear to be underestimated by models, although precise quantifications cannot be made due to the lower resolution.



**Figure 9: Oligocene precipitation data (mm/day) to model comparisons. All proxy data was compared to ensemble means from HadCM3L (UK) and CESM models. (A–I) Precipitation data–model comparisons for three Oligocene time slices. The data is both displayed in a spatial (A, B, D, E, G, H) and zonal mean (C, F, I) of the different precipitations. Differences in A, B, D, E, G and H are calculated as a pointwise difference between the proxy mean value and the model annual mean and then plotted as points of differences. The pink and blue ribbons in C, F and I represent the maximum and minimum differences associated with the zonal means. All proxy data are shown in appendix Table A5.**



## 4.4 Ice sheets

Ice caps expanded during the EOT on Antarctica (Ehrmann and Mackensen, 1992; Shackleton, 1986; Zachos et al., 1992). This ice sheet expansion is supported by benthic foraminiferal isotope trends (Shackleton, 1986; Zachos et al., 1996, 2001b).

By assuming that bottom water temperatures cannot fall below 1°C (Zachos et al., 2001b, 1993, 1996), the majority of the recorded $\delta^{18}O$ shift around the EOT must have resulted, in part, from ice volume changes. Therefore, the benthic foraminifer stable oxygen isotope records suggest that during the EOT the previously small, and most likely ephemeral, Antarctic ice sheet started to grow extensively (Miller et al., 1987). The extent of ice sheet growth on Antarctica during the EOT is not fully understood, but estimates for the Antarctic ice sheet coverage during the Oligocene itself are even more variable. Miller et al.

(1987) suggested that ice volume on Antarctica during the Oligocene was at least 50–70% of the mass of present-day continental ice sheets. Pekar et al. (2006), applying $\delta^{18}O$-to-sea-level calibrations to late Oligocene (26–24 Ma) benthic foraminifera $\delta^{18}O$ records, showed that in the late Oligocene the Antarctic ice sheet was anywhere from 50% up to 125% of present-day ice volume. Similar values were later suggested from paired $\delta^{18}O$ and Mg/Ca analyses on planktic (Lear et al., 2008) and benthic foraminifera (Bohaty et al., 2012).

However, recorded benthic foraminifera oxygen isotope changes at the EOT may be too large to be solely the result of sea ice growth in Antarctica (Coxall et al., 2005), but this depends on the isotopic composition of the formed ice (Bohaty et al., 2012) and the Eocene size of the Antarctic continent (Wilson et al., 2012). Model simulations suggest that significant NH continental glaciation would be preceded by Arctic sea ice (DeConto et al., 2008), making the first occurrence of Arctic sea ice an important issue even though it does not affect global ocean $\delta^{18}O$. A study of sea-ice dependent diatoms and the texture of quartz grains

(St. John, 2008; Stickley et al., 2009) suggested the formation of sea-ice in the Arctic Ocean as early as the middle Eocene (~47Ma). Additionally, ice-rafted debris (IRD) from the late Eocene to early Oligocene (38–30 Ma) suggested the existence of glaciers on Greenland, but it does not necessarily imply continental-scale glaciation (Eldrett et al., 2007; Tripati and Darby, 2018; Tripati et al., 2008). Yet, model simulations (DeConto et al., 2008) render significant glaciation on the NH unlikely under the reconstructed atmospheric $CO_2$ concentrations of the Oligocene. Furthermore, Spray et al. (2019) contradicted the

findings of Stickley et al., (2009), as they found that the quartz grains were transported by fluvial processes and reworked by ocean currents rather than by ice. Collectively, there is inconclusive indication of significant NH continental scale glaciation until the late Miocene or Pliocene.

The Influence of orbital forcing and associated feedback mechanisms Is of great Importance to ice sheet formation. Using a coupled global climate and dynamic ice sheet model, DeConto et al. (2008) and DeConto & Pollard (2003) show that once

atmospheric $CO_2$ falls below ~750 ppm the SH ice caps start to expand rapidly during preferable orbital configuration in a mode and pacing consistent with the deep ocean benthic foraminifer $\delta^{18}O$ records. The onset of EOIS correlates with low obliquity amplitude, and a long 405 ka eccentricity cycle (Pälike et al., 2006a). The influence of eccentricity and obliquity on the Oligocene ice sheets and their sensitivity to orbital variations was acknowledged by numerous studies (e.g., Billups et al., 2002, 2004; Coxall et al., 2005; Flower et al., 1997; Liebrand et al., 2016, 2017; Naish et al., 2001; Paul et al., 2000; Shackleton



et al., 1999, 2000; Wade & Pälike, 2004; Zachos, Pagani, et al., 2001; Zachos, Shackleton, et al., 2001). Glaciations during the Oligocene are mainly driven by eccentricity cycles (1.2 Ma glacial cyclicity) with a higher probability of a glacial interval when obliquity is low as well (Wade and Pälike, 2004). Precession cycles were found to influence Oligocene ice sheets based on deep ocean benthic foraminifer $\delta^{18}O$ (De Vleeschouwer et al., 2017; Liebrand et al., 2016, 2017). Highly symmetrical glacial/interglacial cycles within the mid Oligocene, with a ~10–15 ka phase lag, indicates a direct response of the glaciation

to the eccentricity modulation of precession (De Vleeschouwer et al., 2017; Liebrand et al., 2016, 2017). Only towards the OMT do the cycles become asymmetrical, indicating extended glacial buildup, followed by a rapid retreat, resembling the sawtooth-shaped Pleistocene glacial cycles (Liebrand et al., 2017).

## 5. Flora and Faunal changes

The transition to a colder climate state at the EOT facilitated the formation of stronger latitudinal temperature gradients and

thus the formation of new ecological niches. On land, tropical forests around the mid-latitudes started to shrink and coniferous forests and grasslands expanded (Couvreur et al., 2021; Jaramillo et al., 2006; Kohn et al., 2015; Ma et al., 2012; Salard-Cheboldaeff, 1979; Sun et al., 2014). During the Eocene, tropical areas were extensive; however, the drier and somewhat cooler Oligocene climate substantially reduced tropical areas, with extinctions recorded in the Neotropics (Jaramillo et al., 2006). Woody vegetation cover might also have been reduced during the Oligocene because of the impact on the landscape of

newly evolved megafauna browsers (e.g., *Paraceratherium*) (Sage, 2001). The formation of more arid regions in especially lower latitudes, in combination with lower atmospheric $CO_2$ concentrations, likely drove the expansion of $C_4$ plants (Beerling and Osborne, 2006; Christin et al., 2008; Pagani et al., 2005; Sage, 2001), although the most critical transition for C4 plants expansion didn't take place until the Miocene (Strömberg, 2011).

On top of floral changes, the shift in climate across the EOT lead to one of the biggest invertebrate and mammalian fauna

extinctions within the Cenozoic. The so called 'Grande Coupure' (or 'Grand Coupure de Stehlin') was a turnover event caused by the cooling and drying of NH mid- and high latitudes during the EOT, characterized by the extinction of nearly 60% of all mammal lineages in Europe and the immigration of species from Asia (Escarguel et al., 2008; Solé et al., 2020; Stehlin, 1909). The extinction led to a shift from mainly hyaenodont carnivores to the diversification of carnivoramorphans which include the modern type Carnivora as well as the extinct Viverravidae (Escarguel et al., 2008; Solé et al., 2020). Whereas north-western

China underwent a transition from a perissodactyl (i.e., odd-toed ungulates) dominated fauna in a warm temperate forested area in the Eocene to a small rodent and lagomorph dominant fauna in a forest steppe with a dry temperate climate in the earliest Oligocene (Sun et al., 2014). In contrast to the more dry conditions that prevailed around north-western China during the earliest Oligocene, the area of the upper-Rhein Graben (Germany), still had a flora similar to that of Southeast Asia at around 20° N latitude (Kovar-Eder, 2016). This can also be seen in the fauna, as a wing of a Trogoniformes bird, which are

mostly distributed in tropical to pantropical forests, was found at the Frauenwiler Site in Germany, which dates back to the early Oligocene (30–34 Ma) (Moyle, 2005). Records from the Southern Hemisphere mostly record warm and temperate forest



conditions for the earliest Oligocene (e.g., Amoo et al., 2022; Barreda and Palazzesi, 2021; Hinojosa and Villagrán, 2005). While some records of southern latitudes suggest temperate forest conditions even in the later Oligocene (e.g., Martin, 2006), they also show the slow appearance of more steppe-like, open habitat conditions (Barreda et al., 2010; Barreda and Palazzesi, 625 2021; Palazzesi and Barreda, 2007). This was also recorded in the development and spread of hypsodont (high crowned teeth for increased wear and tear) herbivores after the EOT, in America especially (Flynn et al., 2003; Goin, 2010; Mihlbachler et al., 2011; Marivaux et al., 2020). Meanwhile, during the late Chattian (24.8–24 Ma), the fauna of Europe underwent a change known as "Microbunodon Event" (Scherler et al., 2013). Most likely due to the increased seasonality and drier climate conditions during the 'late Oligocene warming', the spread of ailurids (e.g., red pandas), amphicyonids (also called "bear 630 dogs"), ursids (i.e., bears) and especially the hoofed-mammal communities (e.g., Microbunodon) was facilitated (Mennecart, 2015).

Due to the increased temperature gradients and the appearance of sea ice in the SH (Houben et al., 2013), these oceanic conditions had a great influence on the marine flora and fauna of the Oligocene. Several records have shown strong changes in Southern Ocean food web structure, notably the inception of diatoms as a dominant primary producer (e.g., Lazarus et al., 635 2014; Salamy & Zachos, 1999). This likely impacted the distribution of the whale population, shifting to a filter feeding based feeding strategy (Fordyce, 1980). The stronger seasonality of upwelling of highly nutrient rich waters in the Southern Ocean might have triggered the evolution of migrating mysticete (i.e., baleen) whales (Fordyce, 1980; Marx and Fordyce, 2015). The adaptation to filter feeding enabled mysticete whales to occupy niches which were not yet available to whales before. In the late Oligocene, modern dolphins (which evolved from the Odontocetes) appeared for the first time (Geisler et al., 2011). The 640 evolution of these two cetacean groups have been associated with the glaciation of Antarctica and the resulting water mass structure changes, including seasonal sea ice and increased seasonality in primary production (Houben et al., 2013; Salamy and Zachos, 1999).

## 6. Discussion

The Oligocene provides a natural case study for elucidating the functioning of the earth system under a unipolar icehouse, 645 including its steady-state climate and oceanography as well as its variability and sensitivity to change.

## 6.2 Temperature trends and variability

Our compilation of data shows that not only were the Oligocene oceans warm (O'Brien et al., 2020), but also the sparsely available floral and faunal information from terrestrial settings show pronounced warmth (Fig. 4). On average, the data indicate that the Oligocene was slightly cooler than the Eocene, but much warmer than modern times (Figs 4, 6). Notably, the high 650 latitudes were much warmer than modern. The current dataset suggests only modest tropical warming relative to the present, but it should be noted that the expected ocean warm pools have not yet been sampled. Both sea surface and terrestrial temperature gradients between the tropics and SH midlatitudes are particularly small, which was previously recognized for the



Eocene as well as for the Miocene (e.g., Baatsen et al., 2020; Burls et al., 2021; Hollis et al., 2019; Lunt et al., 2021). For both the Eocene and Oligocene SSTs, this is mainly the result of very high proxy values in the southwest Pacific and the Australo-

Antarctic Gulf (Baatsen et al., 2020; O'Brien et al., 2020). The gradient is very steep beyond 50°S (Fig. 6) while the equator to pole temperature gradient in the NH is more gradual. This means that despite the presence of ice in Antarctica, the mid latitudes of the SH seem to be especially warm in the Oligocene.

Previous work has suggested pronounced warming during the late Oligocene (~26 Ma) (Pälike et al., 2006a; Zachos et al., 2001a). This inference may have been the result of using an older compilation of deep ocean benthic foraminifer $\delta^{18}O$ data,

which showed a sharp >1 ‰ drop at this time (Zachos et al., 2001a). The $\delta^{18}O$ signal was most likely affected by a shift in the geographical origin of the data across this interval. Subsequent single-site high resolution records and compilations showed a highly dampened and more gradual decrease in $\delta^{18}O$ values (De Vleeschouwer et al., 2017; Liebrand et al., 2016; Pälike et al., 2006b). Only few of the compiled records show any evidence for long-term warming throughout the Oligocene. Rather, they show relatively stable values, with some records indicating cooling from the early- to mid-Oligocene, followed by long-term

warming (Fig. 5). On the shorter term, there is strong variability within the data of especially the high- and mid-latitudes (Fig. 5). At this point, the resolution of all records is insufficient to assess if any of this variability corresponds to orbital cyclicity. Similar variability apparent in deep ocean benthic $\delta^{18}O$ records primarily reflects obliquity and therefore high latitude signals (Pälike et al., 2006b; Westerhold et al., 2020). Additionally, stratigraphic constraints on these records are insufficient to assess if this variability is consistent between sites. If the variability was global in nature, its increasing amplitude towards higher

latitudes likely reflects a combination of climatic polar amplification (i.e., Ice/snow albedo and humidity) and/or oceanographic variability (i.e., fronts and upwelling).

When compared to model simulations (Fig. 8 and 9), it is evident that the models show significantly less polar amplification of warming than the proxy-based data. In part, this is likely due to the rather modest warming in the tropics in the proxy data. The dataset is limited as floral data might underestimate temperatures in warmer-than-modern tropical regions (e.g., Huber &

Caballero, 2011). In other parts, climate models are not able to fully reconstruct regional climate variations as closely as proxy data can, and thus probably underestimate regional variability (Laepple et al., 2023). Additional high-quality SST data is necessary to fully evaluate tropical temperatures for the Oligocene. The data also indicates very warm conditions at the higher mid-latitudes and the high latitudes. It should be noted that some of this data is based on $TEX_{86}$, which suffers from large uncertainties in absolute SST reconstructions (see section 4.1.2). However, plant based (NLR) and $U^{k'}_{37}$ derived temperatures

show similar warming as those based on $TEX_{86}$ in the high latitudes, suggesting that there was in fact polar amplification towards the higher latitudes. In addition, floral data and carbonate geochemical records support exceptional mid-to-high latitude warmth during the Eocene (e.g., Creech et al., 2010; Douglas et al., 2014; Willard et al., 2019) which seems to still be the case during the Oligocene. Collectively, the nature of the data might exaggerate the low temperature gradient for the Oligocene. Regardless, our findings are in agreement with those of O'Brien et al. (2020), showing that most of the Oligocene

was similar to the late Eocene greenhouse world, rather than the suggested "icehouse". The model-data comparison highlights the ongoing challenges of fully understanding the complex nature of the Oligocene. Questions remain regarding the formation



of ice in a world with a flattened meridional temperature gradient, when poles were much warmer than today and atmospheric $CO_2$ levels were high (e.g., Baatsen et al., 2020).

## 6.3 Precipitation

Surprisingly, the MAP and DMP show similar values as today (Fig. 7), while based on theory (e.g., Pierrehumbert et al., 2002) one would expect latitudes with dominant low pressure to be wetter and dominant high pressure regions to be drier. Values seem somewhat lower than for the Eocene (Cramwinckel et al., 2022). This drier climate compared to the Eocene is also seen in other terrestrial records (e.g., Couvreur et al., 2021; Dupont-Nivet et al., 2007; Jaramillo et al., 2006; Kohn et al., 2015; Ma et al., 2012; Salard-Cheboldaeff, 1979; Sun et al., 2014) which suggest the expansions of arid regions and reduction of

rainforests. In addition, comparisons to modelled data shows, that models still underestimate especially higher latitude precipitation levels. The model most likely underestimates microclimate development due to the larger grid size which leads to an averaged topography and thus lower predicted precipitation. Whereas the proxy data is biased towards the wetter parts, as there is more plant data available where wetter climates persisted. However, similar to temperature (section 6.1), our understanding of global Oligocene precipitation relies on a limited dataset, mainly sampling Europe (Fig. 3). This scarcity

calls for increased efforts in generating additional Oligocene terrestrial data in the future.

## 7. Conclusions

-   SSTs show no uniform trends throughout the Oligocene but were more variable and on average higher than modern SSTs, especially in mid and high latitudes.
-   Mean annual and coldest month temperatures were generally elevated towards the poles, leaving a weaker temperature
gradient than seen in modern times.
-   MAP/DMP of the Oligocene show similar values compared to modern precipitation, with drier conditions around the equator.
-   Models seem to underestimate the warming around the poles and overestimate the tropics and thereby seem to underestimate polar amplification or equator-to-pole heat distribution. Additionally, precipitation is mostly
underestimated by models, yet no definitive conclusions can be made due to the lack of data available.
-   The Oligocene was not the "icehouse" it was long believed to be and most likely experienced warm global SSTs and air temperatures combined with dried conditions.





## 8. Appendix A

| Locality | Average Age (Ma) | Latitude | Paleolatitude | Number of Taxa | Number of Simulations | Quality Flags | Min MAT (°C) | MAT (°C) | Max MAT (°C) | Min WinT (°C) | WinT (°C) | Max WinT (°C) | Min MinT (°C) | MinT (°C) | Max MinT (°C) | Min MAP (mm/yr) | MAP (mm/yr) | Max MAP (mm/yr) | Min DMP (mm/yr) | DMP (mm/yr) | Max DMP (mm/yr) | References |
|---|---|---|---|---|---|---|---|---|---|---|---|---|---|---|---|---|---|---|---|---|---|---|
| As Pontes basin | 26.1 ±3.7 | 43.45 | 37.62 | 14 | 1 | Taxa assignments dubious, one simulation, macrofloras | 11.2 | 14.8 | 17.7 | 4.0 | 8.9 | 12.1 | -0.8 | 1.9 | 7.0 | 759 | 1000 | 1318 | 20 | 38 | 63 | Cabrera et al. 1995 |
| Belen Fruit & Seed assemblage | 29.25 ±0.75 | -4.75 | -12.62 | 17 | 1 | Taxa assignments reliable, one simulation, macroflora | 23.7 | 25.2 | 25.9 | 22.3 | 23.5 | 25.0 | 15.1 | 16.9 | 18.9 | 1445 | 1738 | 2089 | 11 | 20 | 40 | Manchester et al. 2012 |
| Berwick Quarry | 25 | -38.03 | -47.02 | 33 | 10 | Taxa assignments reliable, high convergence, macro- and microfloras | 13.0 | 15.9 | 18.8 | 8.4 | 11.6 | 14.7 | 3.5 | 6.1 | 9.5 | 1096 | 1754 | 2188 | 44 | 78 | 120 | Pole et al. 1993 |
| Calau Beds | 29.5 ±1.5 | 51.78 | 46.86 | 56 | 10 | Taxa assignments reliable, medium convergence, macrofloras | 11.3 | 15.4 | 19.2 | 3.9 | 8.4 | 13.7 | -1.3 | 2.6 | 6.5 | 912 | 1143 | 1514 | 4 | 24 | 69 | Mai 1998 |
| Cervera (Rasqui quarry, Carulla quarry, Mas Claret, Briançó) | 30.85 ±3.05 | 41.65 | 35.44 | 28 | 10 | Taxa assignments reliable, high convergence, macroflora | 13.6 | 16.3 | 18.9 | 7.6 | 11.2 | 14.2 | 1.7 | 5.5 | 8.8 | 661 | 955 | 1318 | 9 | 24 | 58 | Tosal & Martín-Closas 2016 |
| Cosy Dell | 24.9 ±0.5 | -46.15 | -47.91 | 65 | 10 | Taxa assignments reliable, medium convergence, microfloras | 12.6 | 15.8 | 17.8 | 8.6 | 11.4 | 14.0 | 3.5 | 6.2 | 8.9 | 1202 | 1556 | 1905 | 48 | 74 | 115 | Conran et al. 2014 |
| Daxing'anling | 31.15 ±2.75 | 45.86 | 52.40 | 24 | 10 | Taxa assignments reliable, high convergence, microfloras | 8.0 | 10.3 | 13.6 | -0.4 | 3.0 | 6.0 | -5.3 | -1.5 | 1.5 | 692 | 843 | 1096 | 20 | 32 | 55 | Ma et al. 2012 |
| Daxing'anling 2 | | 46.73 | 53.31 | | | | | | | | | | | | | | | | | | | |
| Guang River | 27.23 | 12.60 | 7.24 | 19 | 1 | Taxa assignments reliable, one simulation, macroflora | 22.8 | 24.3 | 25.5 | 21.2 | 22.4 | 23.9 | 14.3 | 16.9 | 18.4 | 1202 | 1445 | 1738 | 8 | 14 | 29 | Pan 2007 |
| Haselbach Horizon | 29.75 ±0.75 | 51.42 | 46.45 | 32 | 10 | Taxa assignments reliable, medium convergence, macroflora | 12.8 | 15.9 | 18.8 | 4.9 | 8.9 | 13.9 | -0.2 | 3.3 | 7.5 | 871 | 1143 | 1585 | 15 | 33 | 55 | Kunzmann & Walther 2012 |
| Haynes Creek Flora | 30 | 45.00 | 43.22 | 29 | 10 | Taxa assignments reliable, high convergence, macroflora | 8.0 | 11.5 | 13.9 | -0.3 | 3.7 | 7.1 | -5.4 | -0.7 | 2.4 | 692 | 891 | 1148 | 20 | 37 | 63 | Axelrod 1998 |
| Hrazený hill | 29.5 ±1.5 | 50.98 | 46.11 | 32 | 10 | Taxa assignments reliable, low convergence, macroflora | 9.1 | 12.9 | 16.2 | 0.2 | 5.4 | 9.0 | -5.8 | 0.9 | 3.2 | 832 | 1038 | 1380 | 20 | 35 | 66 | Kvaček et al. 2015 |
| Kraskino Flora | 30 | 42.71 | 49.33 | 31 | 10 | Taxa assignments reliable, low convergence, macrofloras | 14.5 | 18.5 | 22.2 | 7.1 | 13.2 | 17.6 | 1.9 | 6.4 | 11.4 | 1096 | 1361 | 1738 | 32 | 43 | 69 | Pavlyutkin 2011 |
| Lea River | 31 ±1 | -41.50 | -53.28 | 10 | 1 | Taxa assignments reliable, one simulation, macroflora | 8.2 | 10.2 | 12.6 | 3.5 | 5.7 | 7.2 | -1.0 | 1.3 | 3.5 | 1000 | 1380 | 1738 | 38 | 63 | 95 | Paull & Hill 2010 |
| Maikop Group | 25 | 40.55 | 39.29 | 30 | 10 | taxa assignments reliable, high convergence, microfloras | 10.1 | 12.8 | 15.6 | 2.6 | 5.3 | 9.1 | -1.9 | 0.8 | 3.3 | 759 | 1005 | 1202 | 23 | 36 | 58 | Popov et al. 2008 |
| Makum Coal Field | 26.2 ±3.2 | 27.25 | 30.59 | 23 | 10 | Taxa assignments dubious, high convergence, macroflora | 23.1 | 24.3 | 26.4 | 21.6 | 23.2 | 24.9 | 16.4 | 18.5 | 20.7 | 1318 | 1667 | 2291 | 7 | 45 | 100 | Awasthi & Mehrotra 1995 |
| Monpeelyata deposit | 23.3 ±0.9 | -41.83 | -49.68 | 43 | 10 | Taxa assignments reliable, medium convergence, microfloras | 10.5 | 13.3 | 15.9 | 5.6 | 8.4 | 11.3 | 0.2 | 3.2 | 6.5 | 794 | 1294 | 1820 | 35 | 63 | 100 | Macphail et al. 1991 |
| Newvale Mine | 24.1 ±1.1 | -46.14 | -47.91 | 99 | 10 | Taxa assignments reliable, low convergence, microfloras | 11.8 | 15.7 | 19.1 | 7.5 | 11.4 | 15.3 | 2.5 | 6.3 | 10.9 | 1096 | 1556 | 2188 | 42 | 84 | 132 | Ferguson et al. 2010 |
| Paleogene basin | 30.45 ±3.45 | 46.35 | 41.54 | 26 | 10 | Taxa assignments reliable, low convergence, macrofloras | 12.8 | 16.2 | 18.9 | 6.2 | 10.2 | 13.9 | 1.1 | 4.0 | 7.6 | 871 | 1107 | 1380 | 17 | 38 | 63 | Erdei et al. 2012 |
| Pitch-Pinnacle flora | 30.95 ±1.95 | 39.12 | 36.56 | 17 | 1 | Taxa assignments reliable, one simulation, macroflora | 10.6 | 12.4 | 13.9 | 3.2 | 5.8 | 7.8 | -1.4 | 0.8 | 2.4 | 832 | 1047 | 1148 | 26 | 38 | 52 | Gregory & McIntosh 1996 |
| Rauenberg | 29.5 ±2.5 | 49.27 | 43.97 | 35 | 10 | Taxa assignments reliable, medium convergence, macrofloras | 12.3 | 15.3 | 18.7 | 5.6 | 9.0 | 14.0 | 0.9 | 4.1 | 8.1 | 794 | 1091 | 1445 | 12 | 30 | 55 | Kovar-Eder 2016 |
| Roudniky area | 31.95 ±1.95 | 50.65 | 45.75 | 33 | 10 | Taxa assignments reliable, medium convergence, macrofloras | 11.2 | 14.1 | 16.8 | 2.8 | 6.9 | 10.6 | -2.6 | 1.7 | 4.7 | 832 | 1102 | 1380 | 20 | 35 | 58 | Kvaček et al. 2014 |
| San Julian Fm | 24 | -49.16 | -54.85 | 18 | 1 | Taxa assignments reliable, one simulation, microfloras | 14.7 | 16.4 | 18 | 9.4 | 10.7 | 12.8 | 2.5 | 4.1 | 6.9 | 912 | 1202 | 1514 | 26 | 42 | 69 | Palazzesi & Barreda 2007 |
| Suletice-Berand | 27.5 ±1.5 | 50.61 | 45.89 | 17 | 1 | Taxa assignments reliable, one simulation, macroflora | 12.6 | 14.5 | 16.1 | 6.0 | 8.7 | 10.3 | 0.5 | 1.8 | 4.4 | 871 | 1096 | 1380 | 28 | 42 | 60 | Kvaček & Walther 1995 |
| Tard Clay1 | 32.9 ±0.9 | 47.50 | 42.73 | 12 | 1 | Taxa assignments reliable, one simulation, macroflora | 14.6 | 16.9 | 19.4 | 8.3 | 11.6 | 14.8 | 2.5 | 7.0 | 8.9 | 759 | 1000 | 1445 | 8 | 13 | 35 | Kvacek et al. 2001 |
| Tard Clay2 | | 47.91 | 42.73 | | | | | | | | | | | | | | | | | | | |
| upper Ruby Basin | 32.9 ±0.7 | 45.11 | 43.10 | 65 | 10 | Taxa assignments reliable, medium convergence, macroflora | 10.2 | 13.5 | 16.8 | 2.7 | 6.0 | 11.0 | -2.8 | 1.3 | 5.4 | 794 | 1028 | 1318 | 18 | 34 | 58 | Becker 1966 |
| Weißelster | 29.75 ±0.75 | 51.42 | 46.45 | 79 | 10 | Taxa assignments reliable, low convergence, macrofloras | 10.0 | 13.9 | 18.9 | 2.7 | 6.6 | 12.3 | -1.7 | 1.5 | 5.7 | 871 | 1138 | 1585 | 20 | 39 | 66 | Gastaldo et al. 1998 |

**Table A1: Results of the nearest living relative (NLR) analysis, showing mean annual temperature (MAT), winter mean temperatures (WinT), mean annual precipitations (MAP) and driest month precipitation (DMP) and their respective minimum and maximum values.**



| SST Proxy | SST calibration details | References |
|---|---|---|
| $U^K_{37}$ | $U^{k'}_{37}$ indices were converted to SST estimates using the global core-top calibration of Müller et al., 1998. | Müller PJ, Kirst G, Ruhland G, Von Storch I, & Rosell-Melé A (1998) Calibration of the alkenone paleotemperature index $U^{K'}_{37}$ based on core-tops from the eastern South Atlantic and the global ocean (60°N-60°S). *Geochimica et Cosmochimica Acta* 62(10):1757–1772. |
| $TEX_{86}$ | TEX86 values were converted to SST using the global logarithmic TEXH86 calibration of Kim et al. (2010). | Kim, J.-H. et al. New indices and calibrations derived from the distribution of crenarchaeal isoprenoid tetraether lipids: implications for past sea surface temperature reconstructions. Geochim. Cosmochim. Acta 74, 4639–4654 (2010). |
| $\Delta_{47}$ | Δ47 SST estimates and sample age were taken directly from the original publications. | Douglas, P. M. J. *et al.* Pronounced zonal heterogeneity in Eocene southern high-latitude sea surface temperatures. *Proceedings of the National Academy of Sciences* **111**, 6582-6587 (2014). Evans, D. *et al.* Eocene greenhouse climate revealed by coupled clumped isotope-Mg/Ca thermometry. *Proceedings of the National Academy of Sciences* **115**, 1174–1179, doi:10.1073/pnas.1714744115 (2018). Petersen, S. & Schrag, D. Antarctic ice growth before and after the Eocene-Oligocene transition: New estimates from clumped isotope paleothermometry. *Paleoceanography and Paleoclimatology* **30**, 1305–1317, doi:10.1002/2014PA002769 (2015). Briard, J. et al. Seawater paleotemperature and paleosalinity evolution in neritic environments of the Mediterranean margin: insights from isotope analysis of bivalve shells. Palaeogeogr. Palaeoclimatol. Palaeoecol. 543, 109582 (2020). |
| $\delta^{18}O$ coccoliths | SST estimates are original published values for small coccoliths with a vital effect correction in Tremlin et al., 2016. | M. Tremblin, M. Hermoso, F. Minoletti, Equatorial heat accumulation as a long-term trigger of permanent Antarctic ice sheets during the Cenozoic. Proc. Natl. Acad. Sci. U.S.A. 113, 11782–11787 (2016). |
| $\delta^{18}O$ planktic foraminifera | Palaeotemperature estimates were generated using the calibration of Kim & O'Neil (1997). | Kim, S. T., & O'Neil, J. R. (1997). Equilibrium and nonequilibrium oxygen isotope effects in synthetic carbonates. *Geochimica et cosmochimica acta*, *61*(16), 3461-3475. |


**Table A2: Summary of the calibrations and references thereof used for the respective sea surface temperature (SST) proxies.**





**23.9 - 23.0 Ma**

| Site | Latitude | Longitude | Event | Average Age (Ma) | Proxy | Number of Data points | Paleolatitude | Lowest Latitude | Highest Latitude | Min Latitudinal error | Max Latitudinal error | Average SST (°C) | Standard deviation (°C) | Analytical error (°C) | Calibration error (°C) | References |
|---|---|---|---|---|---|---|---|---|---|---|---|---|---|---|---|---|
| 269 | -61.68 | 140.07 | OMT | 23.4 | TEX86 | 2 | -57.1 | -59.73 | -54.53 | 2.7 | 2.5 | 16.54 | 0.54 | 1.00 | 2.50 | Evangelinos et al. 2020 |
| 1356 | -63.31 | 136.00 | OMT | 23.4 | TEX86 | 9 | -58.7 | -61.38 | -56.18 | 2.6 | 2.6 | 17.23 | 2.04 | 1.00 | 2.50 | Hartman et al. 2018 |
| 1168 | -42.61 | 144.41 | OMT | 23.4 | TEX86, UK37 | 14 | -50.9 | -53.54 | -48.33 | 2.7 | 2.5 | 25.68 | 1.29 | 1.00 | 2.50 | Guitian & Stoll 2021, Hoem et al. 2022 |
| 1404 | 40.01 | -51.81 | OMT | 23.4 | UK37 | 18 | 34.2 | 31.69 | 36.89 | 2.5 | 2.7 | 25.70 | 0.58 | 0.30 | 1.50 | Liu et al. 2018 |
| 1406 | 40.35 | -51.65 | OMT | 23.4 | UK37, TEX86 | 7 | 34.5 | 32.01 | 37.22 | 2.5 | 2.7 | 26.21 | 1.17 | 1.00 | 2.50 | Guitian et al. 2019 |
| 608 | 42.84 | -23.09 | OMT | 23.4 | TEX86 | 4 | 36.9 | 34.38 | 39.59 | 2.5 | 2.7 | 28.14 | 0.93 | 1.00 | 2.50 | Super et al. 2018 |
| 929 | 5.98 | -43.74 | OMT | 23.4 | TEX86 | 5 | 0.4 | -2.16 | 3.05 | 2.6 | 2.6 | 29.08 | 1.00 | 1.00 | 2.50 | O'Brien et al. 2020, Liu et al. 2009 |
| 516 | -30.28 | -35.29 | OMT | 23.4 | TEX86 | 10 | -35.5 | -38.20 | -33.00 | 2.7 | 2.5 | 29.33 | 0.77 | 1.00 | 2.50 | O'Brien et al. 2020 |

**27.3 - 26.3 Ma**

| Site | Latitude | Longitude | Event | Average Age (Ma) | Proxy | Number of Data points | Paleolatitude | Lowest Latitude | Highest Latitude | Min Latitudinal error | Max Latitudinal error | Average SST (°C) | Standard deviation (°C) | Analytical error (°C) | Calibration error (°C) | References |
|---|---|---|---|---|---|---|---|---|---|---|---|---|---|---|---|---|
| 336 | 63.35 | -7.79 | MOGI | 26.8 | UK37 | 1 | 57.4 | 54.81 | 60.01 | 2.5 | 2.7 | 14.77 | 0.00 | 0.30 | 1.50 | Liu et al. 2009 |
| 511 | -51.00 | -46.97 | MOGI | 26.8 | TEX86 | 1 | -57.0 | -59.67 | -54.47 | 2.7 | 2.5 | 17.54 | 0.00 | 1.00 | 2.50 | Liu et al. 2009, Houben et al. 2019 |
| 516 | -30.28 | -35.29 | MOGI | 26.8 | TEX86 | 6 | -35.8 | -38.51 | -33.31 | 2.7 | 2.5 | 27.87 | 0.41 | 1.00 | 2.50 | O'Brien et al. 2020 |
| 274 | -69.00 | 173.43 | MOGI | 26.8 | TEX86 | 6 | -64.9 | -67.57 | -62.37 | 2.6 | 2.6 | 12.31 | 0.33 | 1.00 | 2.50 | Hoem et al. 2021 |
| 1356 | -63.31 | 136.00 | MOGI | 26.8 | TEX86 | 5 | -58.5 | -61.14 | -55.94 | 2.6 | 2.6 | 19.68 | 1.89 | 1.00 | 2.50 | Hartman et al. 2018 |
| 1168 | -42.61 | 144.41 | MOGI | 26.8 | UK37 | 9 | -52.6 | -55.23 | -50.03 | 2.7 | 2.5 | 23.69 | 1.44 | 0.30 | 1.50 | Guitian & Stoll 2021, Hoem et al. 2022 |
| 1404 | 40.01 | -51.81 | MOGI | 26.8 | UK37 | 8 | 33.7 | 31.15 | 36.36 | 2.5 | 2.7 | 24.95 | 0.48 | 0.30 | 1.50 | Liu et al. 2018 |
| 925 | 4.20 | -43.49 | MOGI | 26.8 | TEX86 | 1 | -1.7 | -4.36 | 0.85 | 2.6 | 2.6 | 25.64 | 0.00 | 1.00 | 2.50 | Liu et al. 2009, Zhang et al. 2013, Inglis et al. 2015, Cramwinckel et al. 2018 |
| 1406 | 40.35 | -51.65 | MOGI | 26.8 | UK37, TEX86 | 4 | 34.0 | 31.48 | 36.68 | 2.5 | 2.7 | 25.86 | 1.15 | 1.00 | 2.50 | Guitian et al. 2019 |
| 929 | 5.98 | -43.74 | MOGI | 26.8 | TEX86 | 17 | 0.0 | -2.59 | 2.62 | 2.6 | 2.6 | 28.72 | 1.08 | 1.00 | 2.50 | O'Brien et al. 2020, Liu et al. 2009 |
| 608 | 42.84 | -23.09 | MOGI | 26.8 | TEX86 | 2 | 36.5 | 33.97 | 39.18 | 2.5 | 2.7 | 30.57 | 1.52 | 1.00 | 2.50 | Super et al. 2018 |
| 690 | -65.16 | 1.20 | MOGI | 26.8 | d18O | 4 | -68.4 | -71.06 | -65.86 | 2.6 | 2.6 | 5.03 | 0.77 | 1.00 | 2.50 | Ehrmann & Mackensen 1992, Gaskell et al. 2022 |

**33.7 - 33.2 Ma**

| Site | Latitude | Longitude | Event | Average Age (Ma) | Proxy | Number of Data points | Paleolatitude | Lowest Latitude | Highest Latitude | Min Latitudinal error | Max Latitudinal error | Average SST (°C) | Standard deviation (°C) | Analytical error (°C) | Calibration error (°C) | References |
|---|---|---|---|---|---|---|---|---|---|---|---|---|---|---|---|---|
| 274 | -69.00 | 173.43 | EOGM | 33.4 | TEX86 | 1 | -64.6 | -67.31 | -61.91 | 2.7 | 2.7 | 11.91 | 0.00 | 1.00 | 2.50 | Hoem et al. 2021 |
| 511 | -51.00 | -46.97 | EOGM | 33.4 | UK37, TEX86 | 13 | -58.0 | -60.77 | -55.36 | 2.8 | 2.6 | 12.94 | 3.29 | 1.00 | 2.50 | Liu et al. 2009, Plancq et al. 2014, Houben et al. 2019 |
| 913 | 75.49 | 6.95 | EOGM | 33.4 | UK37 | 2 | 67.4 | 64.76 | 70.17 | 2.7 | 2.7 | 15.51 | 4.80 | 0.30 | 1.50 | Liu et al. 2009 |
| 1356 | -63.31 | 136.00 | EOGM | 33.4 | TEX86 | 1 | -58.0 | -60.74 | -55.34 | 2.8 | 2.6 | 19.87 | 0.00 | 1.00 | 2.50 | Hartman et al. 2018 |
| 1168 | -42.61 | 144.41 | EOGM | 33.4 | TEX86 | 5 | -55.3 | -58.07 | -52.67 | 2.8 | 2.6 | 24.92 | 0.83 | 1.00 | 2.50 | Hoem et al. 2022 |
| 1404 | 40.01 | -51.81 | EOGM | 33.4 | UK37 | 13 | 32.5 | 29.90 | 35.31 | 2.6 | 2.8 | 26.30 | 0.78 | 0.30 | 1.50 | Liu et al. 2018 |
| 1209 | 32.65 | 158.51 | EOGM | 33.4 | TEX86 | 2 | 29.8 | 27.20 | 32.61 | 2.6 | 2.8 | 26.42 | 0.06 | 1.00 | 2.50 | Kast et al. 2019 |
| 959 | 3.63 | -2.74 | EOGM | 33.4 | TEX86 | 6 | -4.2 | -6.95 | -1.54 | 2.7 | 2.7 | 26.63 | 0.46 | 1.00 | 2.50 | Cramwinckel et al. 2018 |
| 925 | 4.20 | -43.49 | EOGM | 33.4 | TEX86, d18O | 4 | -2.7 | -5.45 | -0.03 | 2.7 | 2.7 | 30.41 | 1.13 | 1.00 | 2.50 | Liu et al. 2009, Zhang et al. 2013, Inglis et al. 2015, Cramwinckel et al. 2018, Tremblin et al. 2016 |
| 744 | -61.58 | 80.60 | EOGM | 33.4 | d18O | 6 | -58.3 | -61.08 | -55.67 | 2.8 | 2.6 | 6.70 | 1.41 | 1.00 | 2.50 | Barron et al. 1991; Gaskell 2022 |

**Table A3: Summary of all compiled sea surface temperatures (SSTs) for all Site locations including the analytical and calibration errors used for each proxy. Top: Available SST data for the Oligocene Miocene Transition (OMT), Middle: Available SST data for the Mid Oligocene Glacial Interval (MOGI), Bottom: Available SST data for the Eocene Oligocene Glacial Maximum (EOGM).**




| 33.9-33 Ma | Age (Ma) | Paleolatitude (33Ma) | Paleolongitude (33Ma) | Proxy | Error (°C) | Mean (°C) | Median (°C) | Lower quartile (°C) | LQ + Error (°C) | Upper quartile (°C) | UQ + Error (°C) | True value Min value (°C) | True value Max value (°C) | True value + Error Min value (°C) | True value + Error Max value (°C) | Reference |
|---|---|---|---|---|---|---|---|---|---|---|---|---|---|---|---|---|
| Cervera | 27.8-33.9 | 38.44 | 2.43 | NLR | 1.8 | 16.3 | 16.1 | 14.5 | 12.7 | 18.0 | 19.7 | 13.6 | 18.9 | 11.8 | 20.7 | Tosal & Martín-Closas 2016 |
| Daxing'anling | 28.4-33.9 | 43.73 | 120.42 | NLR | 2.3 | 10.3 | 10.3 | 8.7 | 6.3 | 13.2 | 15.5 | 8.0 | 13.6 | 5.7 | 15.9 | Ma et al. 2012 |
| IODP1168A | 22.9-34.9 | -62.90 | 149.63 | TEX86 | 0.9 | 25.4 | 25.6 | 25.0 | 24.1 | 26.0 | 26.9 | 23.7 | 26.4 | 22.8 | 27.3 | Hoem et al. 2022 |
| ODP 274 | 24.5-33.7 | -70.24 | 177.16 | TEX86 | 0.7 | 12.4 | 12.4 | 12.1 | 11.4 | 12.6 | 13.3 | 11.9 | 12.8 | 11.3 | 13.5 | Hoem et al. 2021 |
| ODP744 | 24.1-34.9 | -63.68 | 75.30 | δ18O | 1.9 | 7.9 | 8.2 | 6.6 | 4.7 | 8.7 | 10.6 | 4.6 | 11.1 | 2.7 | 13.0 | Barrera & Huber 1991, Gaskell 2022 |
| Paleogene basin | 27-33.9 | 42.08 | 14.10 | NLR | 1.6 | 16.2 | 16.1 | 14.2 | 12.6 | 17.5 | 19.1 | 12.8 | 18.9 | 11.2 | 20.5 | Erdei et al. 2012 |
| Roudníky area | 30-33.9 | 46.27 | 15.68 | NLR | 1.8 | 14.1 | 14.4 | 12.6 | 10.8 | 16.2 | 18.0 | 11.2 | 16.8 | 9.4 | 18.6 | Kvaček & Walther 1995 |
| Tard Clay | 32-33.8 | 42.71 | 19.85 | NLR | 2.4 | 16.9 | 16.9 | 14.6 | 12.2 | 19.4 | 21.8 | 14.6 | 19.4 | 12.2 | 21.8 | Kvaček et al. 2001 |
| Upper Ruby Basin | 32.2-33.6 | 51.56 | -115.06 | NLR | 1.7 | 13.5 | 13.3 | 11.6 | 9.9 | 15.0 | 16.7 | 10.2 | 16.8 | 8.5 | 18.5 | Becker 1966 |

| 33-26.5 Ma | Age (Ma) | Paleolatitude (30Ma) | Paleolongitude (30Ma) | Proxy | Error (°C) | Mean (°C) | Median (°C) | Lower quartile (°C) | LQ + Error (°C) | Upper quartile (°C) | UQ + Error (°C) | True value Min value (°C) | True value Max value (°C) | True value + Error Min value (°C) | True value + Error Max value (°C) | Reference |
|---|---|---|---|---|---|---|---|---|---|---|---|---|---|---|---|---|
| As Pontes basin | 22.4-29.8 | 41.37 | -6.13 | NLR | 3.3 | 14.8 | 14.8 | 11.2 | 7.9 | 17.7 | 21.0 | 11.2 | 17.7 | 7.9 | 21.0 | Cabrera et al. 1994 |
| Belen Fruit & Seed assemblage | 30-28.5 | 0.05 | -84.71 | NLR | 1.1 | 25.2 | 25.2 | 23.7 | 22.6 | 25.9 | 27.0 | 23.7 | 25.9 | 22.6 | 27.0 | Manchester et al. 2012 |
| Calau Beds | 28-31 | 47.80 | 15.65 | NLR | 1.8 | 15.4 | 15.6 | 13.3 | 11.5 | 16.8 | 18.6 | 11.3 | 19.2 | 9.5 | 21.0 | Ferguson et al. 2010 |
| Cervera | 27.8-33.9 | 41.65 | 1.33 | NLR | 1.8 | 16.3 | 16.1 | 14.5 | 12.7 | 18.0 | 19.7 | 13.6 | 18.9 | 11.8 | 20.7 | Tosal & Martín-Closas 2016 |
| Daxing'anling | 28.4-33.9 | 44.03 | 121.21 | NLR | 2.3 | 10.3 | 10.3 | 8.7 | 6.3 | 13.2 | 15.5 | 8.0 | 13.6 | 5.7 | 15.9 | Ma et al. 2012 |
| Guang River | 27.23 | 7.31 | 34.03 | NLR | 1.4 | 24.3 | 24.3 | 22.8 | 21.4 | 25.5 | 26.9 | 22.8 | 25.5 | 21.4 | 26.9 | Pan 2007 |
| Haselbach Horizon | 29-30.5 | 47.48 | 14.95 | NLR | 1.5 | 15.9 | 15.6 | 14.1 | 12.6 | 17.1 | 18.7 | 12.8 | 18.8 | 11.3 | 20.3 | Kunzmann & Walther 2012 |
| Haynes Creek Flora | 30 | 50.89 | -116.83 | NLR | 1.9 | 11.5 | 11.4 | 9.6 | 7.7 | 13.3 | 15.2 | 8.0 | 13.9 | 6.1 | 15.8 | Axelrod 1998 |
| Hrazený hill | 28-31 | 46.94 | 16.09 | NLR | 1.9 | 12.9 | 12.9 | 11.7 | 9.8 | 15.3 | 17.2 | 9.1 | 16.2 | 7.2 | 18.1 | Kvaček et al. 2015 |
| IODP1168A | 22.9-34.9 | -60.95 | 148.67 | TEX86 | 1.4 | 25.3 | 25.4 | 24.6 | 23.3 | 26.4 | 27.7 | 21.9 | 27.9 | 20.5 | 29.3 | Hoem et al. 2022 |
| IODP1168A | 22.9-29.2 | -60.95 | 148.67 | UK37 | 1.5 | 21.3 | 21.1 | 20.4 | 18.9 | 22.1 | 23.6 | 19.2 | 23.4 | 17.7 | 24.9 | Guitian & Stoll, 2021 |
| Kraskino Flora | 30 | 40.09 | 122.70 | NLR | 1.6 | 18.5 | 18.8 | 16.9 | 15.3 | 20.2 | 21.9 | 14.5 | 22.2 | 12.9 | 23.8 | Pavlyutkin 2011 |
| Lea River | 30-32 | -59.66 | 149.94 | NLR | 2.2 | 10.2 | 10.2 | 8.2 | 6.0 | 12.6 | 14.8 | 8.2 | 12.6 | 6.0 | 14.8 | Paull & Hill 2010 |
| Makum Coal Field | 23-29.4 | 24.42 | 92.74 | NLR | 1.2 | 24.3 | 23.9 | 23.5 | 22.3 | 25.8 | 27.0 | 23.1 | 26.4 | 21.9 | 27.6 | Awasthi & Mehrotra 1995 |
| ODP 274 | 24.5-33.7 | -70.03 | 176.95 | TEX86 | 1.2 | 13.5 | 13.3 | 12.5 | 11.3 | 14.6 | 15.8 | 11.4 | 15.7 | 10.2 | 17.0 | Hoem et al. 2021 |
| ODP690 | 24.2-30.1 | -64.25 | -5.70 | δ18O | 1.2 | 4.6 | 4.5 | 3.7 | 2.6 | 5.6 | 6.7 | 3.1 | 6.2 | 2.0 | 7.4 | Mackensen & Ehrmann 1992, Gaskell 2022 |
| ODP744 | 24.1-34.9 | -63.58 | 75.92 | δ18O | 1.2 | 7.0 | 6.9 | 6.0 | 4.8 | 7.8 | 8.9 | 5.3 | 9.6 | 4.1 | 10.8 | Barrera & Huber 1991, Gaskell 2022 |
| Paleogene basin | 27-33.9 | 42.44 | 14.05 | NLR | 1.6 | 16.2 | 16.1 | 14.2 | 12.6 | 17.5 | 19.1 | 12.8 | 18.9 | 11.2 | 20.5 | Erdei et al. 2012 |
| Pitch-Pinnacle flora | 29-32.9 | 44.93 | -108.88 | NLR | 1.7 | 12.4 | 12.4 | 10.6 | 8.9 | 13.9 | 15.6 | 10.6 | 13.9 | 8.9 | 15.6 | Gregory & McIntosh 1996 |
| Rauenberg | 27-32 | 45.68 | 10.51 | NLR | 1.8 | 15.3 | 15.1 | 13.6 | 11.8 | 17.2 | 19.0 | 12.3 | 18.7 | 10.5 | 20.5 | Mai 1998 |
| Roudníky area | 30-33.9 | 46.65 | 15.57 | NLR | 1.8 | 14.1 | 14.4 | 12.6 | 10.8 | 16.2 | 18.0 | 11.2 | 16.8 | 9.4 | 18.6 | Kvaček & Walther 1995 |
| Suletice-Berand | 26-29 | 46.59 | 15.82 | NLR | 1.8 | 14.5 | 14.5 | 12.6 | 10.8 | 16.1 | 17.9 | 12.6 | 16.1 | 10.8 | 17.9 | Cabrera et al. 1994 |
| Tard Clay | 32-33.8 | 43.17 | 19.82 | NLR | 2.4 | 16.9 | 16.9 | 14.6 | 12.2 | 19.4 | 21.8 | 14.6 | 19.4 | 12.2 | 21.8 | Kvaček et al. 2001 |
| Upper Ruby Basin | 32.2-33.6 | 50.98 | -114.80 | NLR | 1.7 | 13.5 | 13.3 | 11.6 | 9.9 | 15.0 | 16.7 | 10.2 | 16.8 | 8.5 | 18.5 | Becker 1966 |
| Weisselster | 29-30.5 | 47.48 | 14.95 | NLR | 1.7 | 13.9 | 13.5 | 12.4 | 10.7 | 15.6 | 17.3 | 10.0 | 18.9 | 8.3 | 20.6 | Gastaldo et al. 1998 |

| 25-23 Ma | Age (Ma) | Paleolatitude (24Ma) | Paleolongitude (24Ma) | Proxy | Error (°C) | Mean (°C) | Median (°C) | Lower quartile (°C) | LQ + Error (°C) | Upper quartile (°C) | UQ + Error (°C) | True value Min value (°C) | True value Max value (°C) | True value + Error Min value (°C) | True value + Error Max value (°C) | Reference |
|---|---|---|---|---|---|---|---|---|---|---|---|---|---|---|---|---|
| As Pontes basin | 22.4-29.8 | 41.80 | -6.44 | NLR | 3.3 | 14.8 | 14.8 | 11.2 | 7.9 | 17.7 | 21.0 | 11.2 | 17.7 | 7.9 | 21.0 | Cabrera et al. 1994 |
| Berwick Quarry | 25 | -52.34 | 147.39 | NLR | 1.2 | 15.9 | 15.6 | 14.5 | 13.3 | 16.7 | 17.9 | 13.0 | 18.8 | 11.8 | 20.0 | Gastaldo et al. 1998 |
| Cosy Dell | 24.4-25.4 | -44.80 | 170.69 | NLR | 1.3 | 15.7 | 15.7 | 14.3 | 13.0 | 16.8 | 18.1 | 12.6 | 17.8 | 11.3 | 19.1 | Conran et al. 2014 |
| IODP1168A | 22.9-34.9 | -57.00 | 147.14 | TEX86 | 2.3 | 24.6 | 25.2 | 23.0 | 20.7 | 26.4 | 28.7 | 18.5 | 27.9 | 16.2 | 30.3 | Hoem et al. 2022 |
| IODP1168A | 22.9-29.2 | -57.00 | 147.14 | UK37 | 1.7 | 25.3 | 25.7 | 24.2 | 22.5 | 26.3 | 28.0 | 22.6 | 28.1 | 20.9 | 29.8 | Guitian & Stoll, 2021 |
| Maikop Group | 25Ma | 35.96 | 47.13 | NLR | 1.7 | 12.8 | 12.8 | 10.9 | 9.2 | 14.2 | 15.9 | 10.1 | 15.6 | 8.4 | 17.3 | Popov et al. 2008 |
| Makum Coal Field | 23-29.4 | 25.46 | 93.70 | NLR | 1.2 | 24.3 | 23.9 | 23.5 | 22.3 | 25.8 | 27.0 | 23.1 | 26.4 | 21.9 | 27.6 | Awasthi & Mehrotra 1995 |
| Monpeelyata deposit | 22.4-24.2 | -55.94 | 149.71 | NLR | 1.9 | 13.3 | 13.3 | 11.1 | 9.1 | 14.9 | 16.8 | 10.5 | 15.9 | 8.6 | 17.8 | Macphail et al. 1991 |
| Newvale Mine | 23-25.2 | -51.28 | -174.24 | NLR | 1.3 | 15.7 | 15.9 | 14.4 | 13.1 | 17.0 | 18.4 | 11.8 | 19.1 | 10.5 | 20.4 | Conran et al. 2014 |
| ODP 274 | 24.5-33.7 | -69.65 | 176.37 | TEX86 | 0.8 | 12.7 | 12.8 | 12.2 | 11.4 | 13.2 | 14.1 | 11.6 | 13.3 | 10.8 | 14.2 | Hoem et al. 2021 |
| ODP744 | 24.1-34.9 | -63.36 | 77.13 | δ18O | 3.3 | 7.9 | 6.3 | 6.1 | 2.8 | 8.1 | 11.4 | 6.1 | 12.8 | 2.8 | 16.0 | Barrera & Huber 1991, Gaskell 2022 |
| San Julian Fm | 24 | -45.93 | -74.05 | NLR | 1.7 | 16.4 | 16.4 | 14.7 | 13.0 | 18.0 | 19.7 | 14.7 | 18.0 | 13.0 | 19.7 | Palazzesi & Barreda 2007 |

**Table A4: All sea surface temperature (SST) data per Site location that was added to the O'Brien et al. 2020 Data-Model comparison. Including standard deviations and lower quartile (LQ) and upper quartile (UQ) errors. Top: All available SST data between 33.9 - 33 Ma; Middle: All available SST data between 22 – 26.5 Ma, Bottom: All available SST data between 25 – 23 Ma.**






| **33.9-33 Ma** | | | | | | | | | | | | True value | | True value + Error | | |
| Locality/Sample ID | Age (Ma) | Paleolatitude (33Ma) | Paleolongitude (33Ma) | Proxy | Error (mm/yr) | Mean (mm/yr) | Median (mm/yr) | Lower quartile (mm/yr) | LQ + Error (mm/yr) | Upper quartile (mm/yr) | UQ + Error (mm/yr) | Min value (mm/yr) | Max value (mm/yr) | Min value (mm/yr) | Max value (mm/yr) | Reference |
|---|---|---|---|---|---|---|---|---|---|---|---|---|---|---|---|---|
| Cervera | 27.8-33.9 | 38.44 | 2.43 | NLR | 196.6 | 955 | 933 | 759 | 562.0 | 1148 | 1344.7 | 660.7 | 1318.3 | 464.1 | 1514.8 | Tosal & Martin-Closas 2016 |
| Daxing'anling | 28.4-33.9 | 43.73 | 120.42 | NLR | 174.6 | 843 | 832 | 714 | 539.9 | 1057 | 1231.4 | 691.8 | 1096.5 | 517.2 | 1271.1 | Ma et al. 2012 |
| Paleogene basin | 27-33.9 | 42.08 | 14.10 | NLR | 212.7 | 1107 | 1096 | 920 | 707.7 | 1343 | 1555.5 | 871.0 | 1380.4 | 658.3 | 1593.1 | Erdei et al. 2012 |
| Roudniky area | 30-33.9 | 46.27 | 15.68 | NLR | 200.7 | 1102 | 1072 | 912 | 711.3 | 1306 | 1506.9 | 831.8 | 1380.4 | 631.0 | 1581.1 | Kvaček & Walther 1995 |
| Tard Clay | 32-33.8 | 42.71 | 19.85 | NLR | 348.4 | 1000 | 1000 | 759 | 410.1 | 1445 | 1793.9 | 758.6 | 1445.4 | 410.1 | 1793.9 | Kvaček et al. 2001 |
| Upper Ruby Basin | 32.2-33.6 | 51.56 | -115.06 | NLR | 169.0 | 1028 | 1023 | 879 | 710.0 | 1213 | 1382.4 | 794.3 | 1318.3 | 625.3 | 1487.3 | Becker 1966 |
| **33-26.5 Ma** | | | | | | | | | | | | True value | | True value + Error | | |
| Locality/Sample ID | Age (Ma) | Paleolatitude (30Ma) | Paleolongitude (30Ma) | Proxy | Error (mm/yr) | Mean (mm/yr) | Median (mm/yr) | Lower quartile (mm/yr) | LQ + Error (mm/yr) | Upper quartile (mm/yr) | UQ + Error (mm/yr) | Min value (mm/yr) | Max value (mm/yr) | Min value (mm/yr) | Max value (mm/yr) | Reference |
| As Pontes basin | 22.4-29.8 | 41.37 | -6.13 | NLR | 281 | 1000 | 1000 | 759 | 478 | 1318 | 1599 | 759 | 1318 | 477.9 | 1599.0 | Cabrera et al. 1994 |
| Belen Fruit & Seed assemblage | 30-28.5 | 0.05 | -84.71 | NLR | 322.4 | 1738 | 1738 | 1445 | 1123 | 2089 | 2411.7 | 1445.4 | 2089.3 | 1123.1 | 2411.7 | Manchester et al. 2012 |
| Calau Beds | 28-31 | 47.80 | 15.65 | NLR | 195.4 | 1143 | 1122 | 973 | 777 | 1355 | 1550.6 | 912.0 | 1513.6 | 716.6 | 1708.9 | Ferguson et al. 2010 |
| Cervera | 27.8-33.9 | 41.65 | 1.33 | NLR | 196.6 | 955 | 933 | 759 | 562 | 1148 | 1344.7 | 660.7 | 1318.3 | 464.1 | 1514.8 | Tosal & Martin-Closas 2016 |
| Daxing'anling | 28.4-33.9 | 44.03 | 121.21 | NLR | 174.6 | 843 | 832 | 714 | 540 | 1057 | 1231.4 | 691.8 | 1096.5 | 517.2 | 1271.1 | Ma et al. 2012 |
| Guang River | 27.23 | 7.31 | 34.03 | NLR | 268.1 | 1445 | 1445 | 1202 | 934 | 1738 | 2005.9 | 1202.3 | 1737.8 | 934.1 | 2005.9 | Pan 2007 |
| Haselbach Horizon | 29-30.5 | 47.48 | 14.95 | NLR | 209.9 | 1143 | 1175 | 973 | 763 | 1387 | 1596.6 | 871.0 | 1584.9 | 661.1 | 1794.8 | Kunzmann & Walther 2012 |
| Haynes Creek Flora | 30 | 50.89 | -116.83 | NLR | 170.6 | 891 | 912 | 755 | 584 | 1091 | 1262.1 | 691.8 | 1148.2 | 521.2 | 1318.8 | Axelrod 1998 |
| Hrazený hill | 28-31 | 46.94 | 16.09 | NLR | 185.4 | 1038 | 1000 | 875 | 690 | 1242 | 1427.1 | 831.8 | 1380.4 | 646.4 | 1565.8 | Kvaček et al. 2015 |
| Kraskino Flora | 30 | 40.09 | 122.70 | NLR | 214.0 | 1361 | 1380 | 1159 | 945 | 1585 | 1798.9 | 1096.5 | 1737.8 | 882.5 | 1951.8 | Pavlyutkin 2011 |
| Lea River | 30-32 | -59.66 | 149.94 | NLR | 369.0 | 1380 | 1380 | 1000 | 631 | 1738 | 2106.8 | 1000.0 | 1737.8 | 631.0 | 2106.8 | Paull & Hill 2010 |
| Makum Coal Field | 23-29.4 | 24.42 | 92.74 | NLR | 331 | 1667 | 1660 | 1406 | 1075 | 2061 | 2391.8 | 1318.3 | 2290.9 | 987.1 | 2622.1 | Awasthi & Mehrotra 1995 |
| Paleogene basin | 27-33.9 | 42.44 | 14.05 | NLR | 212.7 | 1107 | 1096 | 920 | 708 | 1343 | 1555.5 | 871.0 | 1380.4 | 658.3 | 1593.1 | Erdei et al. 2012 |
| Pitch-Pinnacle flora | 29-32.9 | 44.93 | -108.88 | NLR | 161.6 | 1047 | 1047 | 832 | 670 | 1148 | 1309.8 | 831.8 | 1148.2 | 670.2 | 1309.8 | Gregory & McIntosh 1996 |
| Rauenberg | 27-32 | 45.68 | 10.51 | NLR | 222.9 | 1091 | 1072 | 871 | 648 | 1300 | 1523.0 | 794.3 | 1445.4 | 571.5 | 1668.3 | Mai 1998 |
| Roudniky area | 30-33.9 | 46.65 | 15.57 | NLR | 200.7 | 1102 | 1072 | 912 | 711 | 1306 | 1506.9 | 831.8 | 1380.4 | 631.0 | 1581.1 | Kvaček & Walther 1995 |
| Suletice-Berand | 26-29 | 46.59 | 15.82 | NLR | 255.3 | 1096 | 1096 | 871 | 616 | 1380 | 1635.7 | 871.0 | 1380.4 | 615.7 | 1635.7 | Cabrera et al. 1994 |
| Tard Clay | 32-33.8 | 43.17 | 19.82 | NLR | 348.4 | 1000 | 1000 | 759 | 410 | 1445 | 1793.9 | 758.6 | 1445.4 | 410.1 | 1793.9 | Kvaček et al. 2001 |
| Upper Ruby Basin | 32.2-33.6 | 50.98 | -114.8 | NLR | 169.0 | 1028 | 1023 | 879 | 710 | 1213 | 1382.4 | 794.3 | 1318.3 | 625.3 | 1487.3 | Becker 1966 |
| Weisselster | 29-30.5 | 47.48 | 14.95 | NLR | 181.3 | 1138 | 1096 | 955 | 774 | 1312 | 1493.5 | 871.0 | 1584.9 | 689.7 | 1766.2 | Gastaldo et al. 1998 |
| **25-23 Ma** | | | | | | | | | | | | True value | | True value + Error | | |
| Locality/Sample ID | Age (Ma) | Paleolatitude (24Ma) | Paleolongitude (24Ma) | Proxy | Error (mm/yr) | Mean (mm/yr) | Median (mm/yr) | Lower quartile (mm/yr) | LQ + Error (mm/yr) | Upper quartile (mm/yr) | UQ + Error (mm/yr) | Min value (mm/yr) | Max value (mm/yr) | Min value (mm/yr) | Max value (mm/yr) | Reference |
| As Pontes basin | 22.4-29.8 | 41.80 | -6.44 | NLR | 281 | 1000 | 1000 | 759 | 478 | 1318 | 1599.0 | 758.6 | 1318.3 | 477.9 | 1599.0 | Cabrera et al. 1994 |
| Berwick Quarry | 25 | -52.34 | 147.39 | NLR | 310 | 1754 | 1820 | 1387 | 1077 | 1995 | 2305.2 | 1096.5 | 2187.8 | 786.5 | 2497.7 | Gastaldo et al. 1998 |
| Cosy Dell | 24.4-25.4 | -44.80 | 170.69 | NLR | 255 | 1556 | 1549 | 1288 | 1034 | 1786 | 2041.0 | 1202.3 | 1905.5 | 947.7 | 2160.0 | Conran et al. 2014 |
| Maikop Group | 25Ma | 35.96 | 47.13 | NLR | 173 | 1005 | 1000 | 843 | 670 | 1186 | 1358.7 | 758.6 | 1202.3 | 585.6 | 1375.2 | Popov et al. 2008 |
| Makum Coal Field | 23-29.4 | 25.46 | 93.70 | NLR | 331 | 1667 | 1660 | 1406 | 1075 | 2061 | 2391.8 | 1318.3 | 2290.9 | 987.1 | 2622.1 | Awasthi & Mehrotra 1995 |
| Monpeelyata deposit | 22.4-24.2 | -55.94 | 149.71 | NLR | 274 | 1294 | 1349 | 1038 | 763 | 1585 | 1859.1 | 794.3 | 1819.7 | 520.1 | 2093.9 | Macphail et al. 1991 |
| Newvale Mine | 23-25.2 | -51.28 | -174.24 | NLR | 258 | 1556 | 1585 | 1324 | 1066 | 1837 | 2094.4 | 1096.5 | 2187.8 | 838.6 | 2445.6 | Conran et al. 2014 |
| San Julian Fm | 24 | -45.93 | -74.05 | NLR | 301 | 1202 | 1202 | 912 | 611 | 1514 | 1814.4 | 912.0 | 1513.6 | 611.2 | 1814.4 | Palazzesi & Barreda 2007 |

**Table A5: All precipitation (MAP) data per Site location that was used in the Data-Model comparison. Including standard deviations and lower quartile (LQ) and upper quartile (UQ) errors. Top: All available MAP data between 33.9 -33 Ma; Middle: All available MAP data between 22 – 26.5 Ma, Bottom: All available MAP data between 25 – 23 Ma.**


## 9. Code Availability

All scripts and programs can be accessed via DOI: 10.5281/zenodo.10144091

## 10. Data Availability

All supplementary data is available via DOI: 10.5281/zenodo.10143889

## 11. Author Contribution

The scripts and programs for the precipitation model were written by Dr. Xiaoqing Liu, the scripts and programs for the sea surface temperature data were written by Prof. Dr. Matthew Huber. Dr. Charlotte O'Brien provided the sea surface temperature



compilation data. Dr. Tammo Reichgelt ran the nearest living analysis on compiled fossil plant remains. Prof. Dr. Appy Sluijs and Dr. Peter Bijl reviewed the paper.

**12. Competing Interests**

Prof. Dr. Appy Sluijs is an editor at CP.

**13. Acknowledgements**

This work was carried out under the program of the Netherlands Earth System Science Centre (NESSC), financially supported by the Ministry of Education, Culture and Science (OCW).

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
