# Peer review of "Climate variability, heat distribution and polar amplification in the unipolar 'doubthouse' of the Oligocene"

_EGUsphere, 2023_

## Author Comment (AC1)

Dear Editor,

we kindly thank reviewer #1 for their constructive feedback on our paper. Below we respond to all their comments and indicate how we intend to adapt our paper based on them.

Sincerely, also on behalf of my co-authors,

Dominique Jenny

**Review of "Climate variability, heat distribution and polar amplification in the unipolar 'doubthouse' of the Oligocene"**

**General comments:**

The purpose of the paper is confusing. It wasn't clear from the start that it is a literature review and interpretation. The paper also tries to do too much by reviewing the chronostratigraphy, the geography, model-data comparison, fauna and flora, introducing new data using NRL, etc., etc. This could be mitigated by stating earlier (title, abstract, intro) and more clearly the purpose and structure of the paper and by restructuring sections of the paper (see below). The discussion/conclusion sections are a little disappointing. The discussion section doesn't offer any insights into the mysteries or speculation about the remaining Oligocene mysteries (e.g. model-data mismatch, CO2-climate mismatch). The conclusion section was obviously a last-minute addition and needs to be redone. Critiques aside, I appreciated the thorough overview and learned some new things about the Oligocene. This could be a nice addition with some additional work.

Reply: We agree that the paper needs to be restructured to be more on point, and more in line with what the title and abstract promises. Also in concert with the advice of Reviewer #2, will remove the sections that aren't directly crucial to understand the data review we present (e.g. chronostratigraphy, fauna and flora). In line with this, we will also revisit the discussion and will re-write the conclusion section.

**Title:**

The title is misleading. This is a review paper and the title should reflect that.

Reply: We will change the title to: "Climate variability, heat distribution and hydrology in the unipolar icehouse of the Oligocene"; a review and data-model comparison" thereby also incorporating a comment by Reviewer #2.

**0 Abstract:**

Page 2, Line 14-15 – I don't understand this sentence. Wouldn't the Oligocene be a good analogue for a future climate state with unipolar glaciation?

Reply: We will restructure the beginning of the Abstract a bit to make it clearer. Indeed, the Oligocene is a useful analogue, yet we call it imperfect as several boundary conditions, such as continental geography and ice volume aren't exactly the same.

Page 2, Line 28 – Delete "while still maintaining a unipolar icehouse state."

Reply: This will be removed

**1 Introduction:**

Most of the paper is well written. The Introduction is the exception and requires careful editing.

Page 3, Line 4 – "equilibrate" is used incorrectly here.

Reply: The wording will be changed here

Page 3, Line 43-44 -  Awkward sentence. Please fix.

Reply: Sentence will be fixed

Page 3, Line 51-52 -  Awkward phrasing. Please fix.

Reply: Sentence will be fixed

Page 4, Line 80 - I'm not sure why whales are mentioned here.

Reply: The sentence here will be adjusted to reflect the importance of such an evolution in the context of the sentence.

Page 5, Figure 1 caption - Maybe the colors didn't transfer correctly, but some of the colors and shapes mentioned in the caption don't match the colors in the legend of the figure (for (b)). Misspelling of "Transition" as well.

Reply: The figure caption will be corrected and the missing methods will be added to the captions.

Page 6, Line 94 – The authors mention variability in continental ice volume paced by eccentricity and obliquity. The authors should follow that with an explanation of why/if that's significant and how it might be seen in the proxy records (given that they are low resolution).

Reply: An explanation will be added here about the significance of the cyclicity of the record, and where it can be seen.

Page 6, Line 99 – "Here.." Should be start of a new paragraph.

Reply: A new paragraph will be made here.

Page 6, Line 100 – "after a chronostratigraphic section" Awkward phrasing.

Reply: The section between line 99 and 104 will be extended and rephrased to include more detail on the paper's purpose, also considering the above remark entitled 'General Comments' by the reviewer.

Page 6, Lines 99-104 - At the end of the introduction, the authors finally explain that the purpose of this paper is to review the current state of knowledge regarding the Oligocene climate and thus provide constraints on boundary conditions, compile proxy data, and identify points of interest. This point should be made much sooner so that readers understand the purpose of the paper.

Reply: This will be considered in the re-shaping of the introduction.

Page 6, Lines 99-104 - There is no mention of numerical climate simulations as in the abstract. Model-data comparisons are made later in the paper, and should be mentioned here. There is also no mention of constraining ocean circulation or ice dynamics though a large part of the introduction discusses these. The authors need to make clearer how climate simulations are used in this paper with some quick details on them, if anything on the ocean or ice state of the Oligocene is explored further, and potentially more details on the proxy records used and for what reason? There is not much on the purpose of the paper and it's all at the very end of the introduction - they could expand on what the paper will discuss and could mention the main purpose it serves sooner in the introduction.

Reply: We agree with this comment, hence the section between line 99 and 104 will be extended and rephrased to include more detail on the paper's purpose. Especially sections about the modelling will be added and the removed sections will be cut from this section.

**2 Oligocene chronostratigraphy and event nomenclature:**

This section should be removed completely from the paper or moved to Supplemental Data.

Reply: Most of this section will be removed following the comments by Reviewer #2 and only the sections about relevant nomenclature and isotopic stages will be kept.

Page 7, Line 112 – "GGSP" should be "GSSP", right?

Reply: Indeed, this typo will be corrected

Page 7, Line 116 - The OMB is stated as 23.04 Ma here and 23.03 Ma in the rest of the sections

Reply: This mistake in the text will be corrected and the sentence will be re-written to be clearer that this date does not mean the OMT.

Page 8, Line 134 – Spell out 'GPTS' as 'geomagnetic polarity time scale (GPTS)'

Reply: This will be added.

Page 9, Line 177 - Why does the last zone end at 23.07 Ma when the Oligocene ends at 23.03 or 23.04 Ma?

Reply: This section will be removed entirely from the paper, as mentioned above.

Page 10, Line 196, 198, 201 - Why is there a question mark after 'Axoprunum'?

Reply: This section will be removed entirely from the paper, as mentioned above.

**3 Boundary conditions for Oligocene climate:**

Page 11, Line 236 – "boundary conditions" is a modeling term and inappropriate here. Replace with "geography" or "continental position"

Reply: This terminology will be changed

Page 12, Figure 3 caption – Add an elevation scale bar. Also 'NRL' is supposed to be 'NLR'.

Reply: The NLR will be corrected. We will contact the maker of this map for an elevation scale bar. Importantly, however, paleoelevation is very poorly constrained for most of the globe during the Oligocene, which will be included in the caption.

Page 13, Line 266 – "strongly affected regional ocean circulation…" Here the authors are repeating a hypothesis that is by no means certain and should equivocate appropriately "may have strongly affected". They do this throughout the paper, stating hypotheses, sometimes model results, as fact or certainty. Please clarify appropriately.

Reply: The phrasing here will be adjusted, and the paper will further be checked for more hypotheses that have been worded as statements.

Page 15, Line 348 – The authors mention an increase in ocean methane hydrates, peat, and wetlands as a result of colder temperatures and ice expansion. They could add a sentence that explains how the expansion of those environments are a consequence of colder temperatures and if there is evidence of these expanding during the Oligocene or if it is just a probable guess given the cooling climate.
Reply: This section will be removed entirely from the paper, also following the advice of Reviewer #2.

Page 15, Line 358-380 – How does these CO2 levels/trends compare with the new CO2 from Honisch et al., Science?

Reply: The paper of Honisch et al (Cenozoic CO2 Proxy Integration Project (CenCO2PIP) Consortium) will be added to the CO2 section. This paper was published after submission of this paper.

Page 16, Line 377 - "16esults" typo

Reply: The typo will be corrected

**4 Climate proxy data:**

Page 16, Line 385 - Why did the authors choose fossil plant remains to analyze and why only fossil plant remains (not paleosol carbonates, etc)?

Reply: There are of course many more possible archives to include, some of which were described in the Flora and Fauna section but much of this information is fragmented and poorly dated, which limits its usefulness. For this reason, we focused on vegetation data, which we considered useful for data-model comparison.

Page 17, Line 405-407 – The authors mention an absence of high latitude data (no fossil plant remains suitable for their method). How could this absence affect their results, or how much uncertainty does this add when estimating a meridional temperature gradient for the Oligocene?

Reply: This essentially means we cannot independently assess meridional temperature gradients based on vegetation data, which we hence do not do. We include a recommendation to revisit the cited materials from the Antarctic margin for quantitative analyses, to solve this issue.

Page 17, Line 415 - One data point in the NH mid-latitudes is significantly different from the other mid-latitude data points - where is this NH data point? What sort of biome is it from?

Reply: Upon reviewing that specific datapoint, the Makum Coal Field, we re-evaluated its paleolatitude at ~25 N, making its climate reconstruction consistent with the other tropical floras. We have therefore removed the comment about it being anomalously warm amongst mid-latitudinal temperature reconstructions.

Page 18, Figure 4 – Use the same y-axis for each plot. As it lies now, the winter temperatures appear the same as the MATs until you see the slight difference in the y-axes. Also, in the figure caption, it states the (b) plot PI values are in brown but they appear gray just like in plot (a).

Reply: We will adjust the y axis for the plot and change the figure caption to match the color of the figure.

Page 19, Line 460 - Random parenthesis in statement. Also, the authors could add one more sentence on how pH affects the fractionation of oxygen isotopes in this environment (why an overestimation of 1.5C has to be taken into consideration?).

Reply: The parenthesis will be removed. We will reconsider mentioning the pH influence on the SST reconstructions and minor overestimations.

Page 20, Figure 5 - This is a nice figure, though the shapes are so small I can't really place many of the sites to a line. They could instead do just three different shapes for each type of SST proxy. There are also some SST sites in Figure 3 that aren't in Figure 5 (i.e. Site 913 and others), if they aren't included in Figure 5 then are they still used / needed in Figure 3? Why weren't some used in Figure 5?

Reply: Figure 3 shows all used data sites for all figures presented. Specifically, 913 was only used in Figure 6, and not included in Figure 5 as it did not span the entire Oligocene. To assess long-term trends, we only include Sites that span throughout all of the Oligocene in Figure 5. We do see how this might lead to some confusion and thus will consider including all sites from Figure 6 in Figure 5 as well.

We agree that the figure could be clearer on the symbols and we will try to make the symbols more readable. We will also try to make the Site symbol the same color as the line it is in.

Page 21, Figure 6 - This is a great figure, very concise with a lot of information. It's interesting too that SST seems ~constant from about 40S to 40N for the Oligocene and sort of late Eocene as well, but modern SSTs have a narrower curve peaking from about 20S to 20N. They could mention something on this.

Reply: This is a good point and one that has received attention in recent work. We will add a sentence in the results about the flattened temperature gradient between those latitudes during the Oligocene.

Page 22, Figure 7 - Why reconstruct driest month precipitation and not wettest month precipitation? Along those lines, why reconstruct winter temperatures and not summer temperatures (Figure 4)? Could just include one sentence stating the benefits of understanding winter temperatures and driest month precipitation.

Reply: Thanks for the suggestion. We have included the following sentence in the methods section (L. 392): "These variables were used here as plant distribution is sensitive to average annual conditions, precipitation seasonality, and the temperature of the coldest season, and therefore can be reconstructed with relative confidence."

Page 22, Line 496 – The authors generally state that the Oligocene MAP values are similar to modern day, but the mid-latitude values (for either hemisphere) seem higher than modern values (Figure 7)?

Reply: We agree with this assessment, and will adjust the text and discussion accordingly.

Pages 23 and 24 – The authors don't mention similarities or differences between the models here at all (HadCM3L vs CESM). According to Figure 8, it seems there are a lot of similarities between how the models reconstruct surface temperature compared to the proxy records. It's interesting that you see such similarities between CESM and HadCM3L (though on that - parts C/F/I of Figure 8 could use a darker color for CESM mean temperature difference since the light pink is a bit difficult to see).

Reply: We will add the sentence: "Despite utilizing two very distinct models with different boundary conditions, the temperature discrepancy between the model and data remains similar." at the beginning of this section 4.3.1 Temperature proxy to 515 model comparison, Line 516

We will change the colour of the CESM to a darker colour.

Page 24, Line 533 – The authors say the differences plotted are 'calculated as a pointwise difference between the proxy mean value and the model annual mean'. Two thoughts on this: a) how was the model annual mean calculated (was the annual mean for the corresponding grid cell used or the general region / a group of grid cells) and b) are there any seasonal biases within the proxy records that could be causing greater differences since only annual means are used for the simulations? If not, they could just state somewhere that there are no known seasonal biases for these records.

Reply:

a) The model annual mean value is derived from the nearest grid point to our study site. We will add a sentence about that in the paper (Line 514)

b) All proxies come with potential biases. Vegetation does not actually respond to mean annual temperature but to seasonal extremes. Marine proxies from pelagic organisms may suffer from seasonal growth and export. However, all proxies as used here are calibrated to mean annual temperature with an associated error that principally includes such seasonal biases. To accommodate this comment by the reviewer, we will include this information in the paper.

Pages 25 and 26 - Similar to what I said above, similarities and differences between CESM and HadCM3L could be mentioned, I think the light pink color for CESM in Figure 9 parts C/F/I is too difficult to see, and I'm a bit confused on the 'pointwise difference' method regarding which grid cells were used for the model values. I also think red should correspond with drier (according to data) conditions (negative precipitation difference) and blue should correspond with wetter (according to data) conditions (positive precipitation difference). It's a bit confusing that we are looking at precipitation as a 'data minus model' value where the data usually represents wetter conditions than the model simulates and yet the data points on the plot are in red colors which traditionally represent drier conditions but this is really just a preference.

Reply: The color of the CESM in Figure 9 will be adjusted to make it more visible. Also the color for the precipitation will be adjusted so that green colors will be corresponding to wetter conditions and yellow to drier conditions in order to avoid confusion with the temperature modelling results.

Page 27, Line 583 – See inappropriate capitalizations. "Is of great Importance"

Reply: This section will be removed.

Page 27, Section 4.4 – This section on Ice Sheets is out of place. Unlike the rest of Section 4, it has no original data or interpretation. It's simply a literature review and should be included in Section 3.

Reply: This section will be removed from the paper.

**5 Flora and faunal changes:**

Similar to last comment, why is this section not with the literature review in section 3?

Reply: This section will be removed entirely from the paper.

**6 Discussion:**

Page 31 – The authors mention there's a discrepancy between SH and NH MAP (in Oligocene and not in modern) in the result section (Figure 7). A discussion of why would be welcome here.

Reply: This is a very good observation and definitely needs to be addressed in the future version of the paper. We thus will add that to the precipitation section of the discussion.

Page 31, line 686-687 – The authors conclude that the Oligocene was anomalously warm relative to CO2 levels. They should discuss potential explanations for this apparent mismatch.

Reply: We agree that this mismatch needs to be discussed, and we intend to add a 6.3 where we discuss the next challenges for understanding Oligocene climates, which includes climate sensitivity, polar amplification and the response of the hydrological cycle.

**7 Conclusions:**

It reads as though the authors ran out of energy before the Conclusions. There's no introductory statement and the bulleted list is very sparse. Besides the abstract, the conclusions are the most frequently read section of a paper. The authors don't state that the manuscript is a review or distinguish between their contributions and what's review.

Reply: We will rewrite the conclusions to fit expectations based on the paper.

Page 31, Line 704 - The second conclusion is specific to the proxy records, not so much what the simulations show, so maybe mention that, though I guess they have a conclusion point specific to the simulations below.

Reply: We will make that clearer.

Page 31, Line 706 - I generally agree with the third conclusion, except Oligocene MAP values around the mid-latitudes appear higher than modern MAP mid-latitude values according to Figure 7.

Reply: This will be adjusted.

Page 31, Line 711 - In line with the last conclusion point, the authors could mention that that may be related to the lack/variability/uncertainty in Oligocene CO2 records. CO2 is not well constrained so perhaps the Oligocene is, in part, not the "icehouse" it seemed to be before because CO2 was, in reality, a bit higher than some older records estimated.

Reply: We will add that to the conclusion.

---

## Author Comment (AC2)

Dear Editor,

we kindly thank reviewer #2 for their constructive feedback on our paper. Below we respond to all comments and indicate how we intend to adapt our paper based on them.

Sincerely, also on behalf of my co-authors,

Dominique Jenny

Jenny et al present a review of the Oligocene, an 11-million-year long time interval. The review is concerned with:

(1) a general introduction,

(2) Chronostratigraphy and event nomenclature,

(2.1) Planktonic foraminifera zones,

(2.2) Calcareous nannofossil zones,

(2.3) Radiolarian biostratigraphic zones,

(2.4) Dinoflagellate cyst zones,

(3) Boundary conditions for Oligocene climate,

(3.1) Geographical boundary Conditions,

(3.2) Ocean circulation,

(3.4) Carbon cycle,

(4) Climate Proxy data,

(4.1) Temperature,

(4.1.1) Continental Mean Annual Temperature,

(4.1.2) Sea Surface Temperature,

(4.2) Precipitation,

(4.3) Data-model comparison,

(4.3.1) Temperature proxy to model comparison,

(4.3.2) Precipitation proxy to model comparison,

(4.4) Ice sheets,

(5) Flora and Faunal changes,

(6) Discussion,

(Note, 6.1 is missing).

Reply: Numbering will be corrected

(6.2) Temperature trends and variability,

(6.3) Precipitation, and

(7) Conclusions.

This review constitutes a very ambitious, admirable, and important project, that necessarily needs to make broad-brush summaries and interpretations about the main climatic and oceanographic factors that determined the Oligocene palaeoclimate system. In principle, I am sympathetic to this effort. The paper is well written.

However, I believe that for this paper to become an authoritative reference paper much more work is needed. I am no expert on all topics reviewed here. But on topics on which I do consider myself reasonably knowledgeable I find that they are often marked by too many inaccuracies, some outdated discussion points, and several key omissions. By proxy, I have therefore little faith that the parts of the review concerned with topics on which I am no expert, accurately capture the current state of knowledge. I am left with the impression that this paper 'could have been so much better'.

To me the paper suffers from not having included enough Oligocene workers, many of whom have dedicated their professional lives to understanding topics ranging from age models, marine microfossils (calcareous nannofossils, foraminifera, radiolaria, etc.), macrofossils (animals and plants), ice sheet modelling, palaeogeography, etc. Compare, for example, this Oligocene review to Steinthorsdottir et al., 2021, who consulted 3 to 4 times as many experts for their Miocene review. (Here my criticism is mainly directed at coauthors Bijl, Huber, and Sluijs; all experienced and senior workers, some of whom have been involved in the Steinthorsdottir effort. To me, and as far as I can judge, it seems that with greater input from these authors, also in study design, the manuscript could have been lifted to a higher level already. Jenny's efforts in leading this study, as a PhD student, is admirable to say the least.)

I believe that revising this manuscript will result in a much better paper. To me the authors have two options: Option 1) shorten the manuscript considerably by removing much of the review text that is not pertinent to the temperature/precipitation compilation and model comparison, i.e. what I consider the main strength of this paper, and expertise of the authors, or Option 2) include more expert knowledge, work with the text and develop the discussion to lift this paper to a higher level. I recommend the second option, and I believe it could make, after substantial revisions, for an authoritative Oligocene reference paper. Ultimately, it is the author's choice.

I will provide some below and wish the authors much success with revising this document and planning the next steps ahead. I am sorry that I am not more positive in this instance.

Reply: We agree and choose for the reviewer's Option 1. We will be shortening the manuscript to the sections that are only important to understanding the newly presented compilation of temperature and precipitation data, and data-modelling comparison and only retain the background information that is important for their context. Thus Chapter 2, large parts of Chapter 3, section 4.4 and Chapter 5 will be removed. Relevant information will be included in the introduction or discussion chapters. This makes a much more concise paper, focused on the main issue identified in the reformulated (also considering the comments by Reviewer #1) introduction.

Major comments

1. Structure of the paper: I believe that inclusion of subheadings concerned with magnetostratigraphy, astrochronology, oxygen and carbon isotope stratigraphy would strengthen the paper. Perhaps compare to Steinthorsdottir et al., 2021, to see how they went about their Miocene review. If this comparison is unjustified, please say so in the rebuttal.

Reply: Sections 2.1-2.4 will be removed from the paper.

2. Doubthouse/Unipolar Doubthouse: Most workers use "Doubthouse" to describe the latest Eocene (between the MECO and EOT), where the climate system was cooling and "doubting" to jump from a "Greenhouse" state to a "Unipolar Icehouse" state. I much prefer to reserve the phrase "Doubthouse" for the Late Eocene, and the usage of "Unipolar Icehouse" to describe the state of the Oligocene to Pliocene time interval. As an aside, I also prefer "Unipolar Icehouse" over the phrase "Coolhouse". The usage of "Unipolar Doubthouse" in your title is a mixture of "Unipolar Icehouse" and "Doubthouse" and is confusing to me. I am not entirely sure what is meant by this phrase. Is the system "doubting" to be unipolarly glaciated? Or "doubting" to go back to a "Greenhouse" state, or to a "Bipolar Icehouse" state?

Reply: We will change the title of the paper to: "We will change the title to: "Climate variability, heat distribution and hydrology in the unipolar icehouse of the Oligocene"; a review and data-model comparison" thereby also incorporating a comment by Reviewer #1.

3. Figure 1: This figure has many inaccuracies. Almost all boundaries or events (apart from those around the EOT, which seem to be OK) are indicted in the wrong place. Furthermore, the MOG is not a previously defined event, the MOGI is. The MOGI as drawn does not cover the MOGI as defined in the literature. The OMT does not cover the largest glaciation across the OMB. The Mi-1 points to an d18O low, not a d18O high, and moreover it is best to abandon Mi and Oi nomenclature completely (more about this below).

Reply: We thank the reviewer for spotting this; most likely a last-minute mishit in illustrator. All boundaries will be checked and moved where necessary, and we agree to abandon Mi and Oi points terminology.

4. Line 108/109, the Oligocene time scale: For an Oligocene review paper, it is important to get the age scale right, and to describe it explicitly (not just mentioning it in a figure caption). The absolute ages of the polarity time scale in the GTS2012 (i.e., the GPTS2012) are of greater quality than those incorporated in the GTS2020/GPTS2020, especially for the latest Oligocene where the GTS2020 incorporated erroneous ages. Admittedly, large efforts have been put toward improving biostratigraphic markers and zonations for the GTS2020, but also their absolute ages are unfortunately too often based on an inferior GPTS2020 scale (compared to the GPTS2012). I recommend using GPTS2012 to any Oligocene (age model) worker, and an Oligocene review paper needs to address the errors in the G(P)TS2020, if the authors decide to stick with this age scale as their preferred option.

Reply: For our contribution regarding large-scale Oligocene trends and data-model comparisons of larger time intervals, the choice of time scale is relatively unimportant. We have therefore chosen to retain the 2020 time scale. We will include this information in the methods section and in the data file so that this can always be updated when an improved time scale is available.

5. Line 353 to 380: This CO2 discussion needs more work. What are the implications for radiative forcing? For example, how to explain the Late Oligocene Warming whilst CO2 is decreasing? This is a major topic in Oligocene research, and it is not mentioned or discussed. For this manuscript to become an authoritative review, such a major climatic enigma needs to be addressed, because it will help with formulating new and important research questions.

Reply: This is a good point. We will include a section 6.3 on outstanding issues, including the presumed late Oligocene warming and its potential discrepancy with the CO2 reconstructions. Additionally, we will discuss it in the light of the recently published CO2 record by the Cenozoic CO2 Proxy Integration Project (CenCO2PIP) Consortium.

6. Discussion and Conclusions, both seem quite superficial to me. Can we take a step back, and distil major research questions for the Oligocene?

Reply: The discussion will be reworked to be more pinpointed at the questions posed in the introduction. The conclusion will be reformatted and focused accordingly.

Minor comments

• Line 13: Perhaps start the abstract with why the Oligocene is of any interest to anybody? It may help draw some more readers in.

Reply: We will change the first sentences of the abstract so it becomes more clear what the interest of the Oligocene is.

• Line 14: Not entirely sure why this would lead from the first line of the abstract. To me the gradients could be a good analogue. Ice sheet geometry, palaeogeography, etc, seem to me the limiting factors in the Oligocene being a palaeo analogy for long term future climate states.

Reply: We will rephrase these sentences, in tandem with the revisions made following the previous comment

- Line 23: proxy data instead of proxy based data?

Reply: This will be changed to proxy data

- Line 25: "In line with previous…" Does this refer to Oligocene modelling efforts? Or modelling of warmer climates in the general. I thought that this was a feature of Eocene modelling output but was unaware of this being a problem in general, also for the Oligocene unipolar icehouse. Perhaps clarify?

Reply: We will clarify this sentence that we indeed mean previous proxy-model comparisons.

- Line 23 to 27: should we learn a lesson from these observations? Is it worth making that lesson explicit in the abstract (e.g., much more effort needs to be invested in improving Oligocene climate simulations?)

Reply: We will add a final conclusional sentence to the abstract about the model simulations.

- 1: Position Mi-1 is wrong. In general, it is best to refrain from using Oi and Mi numbering. Oi and Mi numbers were originally defined (in very low-resolution records from the 1980s and early 1990s) as oxygen isotope zones, like biozones, that lasted several hundreds of thousands to millions of years (see original Miller papers and Wright papers). Over the years, these zone numbers were used to refer to the oxygen isotope "highs" (often called "events") at the base of the original oxygen isotope zones, yet this is not how they were originally defined. The arrows in Fig 1, suggest that the authors also use the Oi and Mi numbers to indicate events. If this usage is adopted, which I do not recommend, then please refer to high oxygen isotope values. E.g., the position of the Mi-1 is not correct in this figure (you refer to an oxygen isotope low). Furthermore, even when the Mi-1 (short for "Miocene oxygen isotope zone 1") is indicated correctly in Fig. 1, it would still fall in the latest Oligocene (yet another reason to abandon Oi and Mi numbers). Lastly, nobody has even been able to explain to me what makes the Oi1b, Oi2a, and Oi2b stand out as particularly interesting, or exceptional "events", probably because these numbers were devised as zones originally.

Reply: We will remove the Mi and Oi arrows/ events from the figures and not use those in the figure in general.

- 1: MOGI instead of MOG? Also, this band is not drawn according to the original definition. You may choose to redefine the main intervals of interest, but then they need to be defined in the text.

Reply: We do define the MOGI in section 2 but only with the Chrons, hence we will add the actual ages of those Chrons to the text. We will also rename the MOG in the figure one to MOGI and make sure that in the text it will always be called MOGI.

- Line 90: Oligoene (missing c)

Reply: The typo will be corrected.

- Line 90: MOG, where G stands for "glacial". In MOGI, this stands for Glacial Interval, and refers to a longer period of mean elevated d18O values (punctuated by shorter lasting d18O decreases of about 1 per mil amplitude, interpreted to be Antarctic "interglacials" and/or deep-water warming events). The use of glacial in MOG is a bit strange and makes me think of something more similar to the last Pleistocene glacial cycle.

Reply: We agree and as mentioned with the previous comment, will use MOGI from now on in the paper.

- Line 99: See Major Comment #1

Reply: We will add subheaders within section 2, to make the section more structured for the reader. Also figure 2 was removed together with sections 2.1-2.4

- Line 109: I strongly recommend using G(P)TS2012 over G(P)TS2020. See Major Comment #3.

Reply: We replied to this comment at the above Major Comment #3.

- Line 115: Beddow et al., 2018 did not work on the OM Boundary type section (Lemme Carrosio) but derived an astronomically calibrated age for the base of C6Cn.2n from Pacific sediments (perhaps make that clearer, because the current sentence can be interpreted as if Beddow et al did work on that section). Also, the 100 kyr error is not mentioned by Beddow et al. and seems too large.

Reply: This will be corrected.

- Line 110 to 125: This section would be much strengthened with the addition of a description of astrochronozones and potentially astro-unit-stratotypes. GSSPs are useful to an extent, but have their limitations (see e.g., Hilgen et al., 2020, Newsl. On stratigraphy). Potentially add a magnetostratigraphic description as well

Reply: Based on above Major Comment 0 and the advice of the reviewer, this section will be removed

- Lines 126 to 145: Overall a good summary, however, as stated above, I would shy away from reiterating Oi and Mi numbers. EOT, MOGI, and OMT are more useful terms in my opinion. Potentially in addition with the Late Oligocene Warming (LOW). MOGI definition used here, instead of MOG in figure caption. Please note that the MOGI is defined as "a generally cold but highly unstable mid-Oligocene time interval (∼0 My to 26.3 My ago), which we refer to as the Mid-Oligocene Glacial Interval (MOGI)", not as "a ~1 Ma year phase of profound cooling/glacial expansion". Usage of Ma is erroneous in this context too (Ma refers to an age not a duration). In general, this section needs some work/detailed checking, as I have not checked all statements, and I may have overlooked other potential errors.

Reply: The figure text will be adjusted as mentioned above. In this section the definition of the Mi/Oi will be left in as they have often been used in the literature and thus need mentioning in the text, but we will also explain why the community is moving away from using them. We'll include the description of MOGI as indicated by the reviewer and use ages from Liebrand et al. and of course correct Ma to Myr.

- Section 2.1. The GTS2020 is abandoned here?

Reply: This section will be removed.

- Section 2.2. Similar comment. Agnini et al., 2014, cannot have used GTS2020. Please mention on which age scale these bioevents are given.

Reply: This section will be removed.

- Section 2.3. Same here.

Reply: This section will be removed.

- Section 2.4. and here. For Chapter 2, in general, best to refer to an accepted general age scale. I recommend G(P)TS2012. Where biostratigraphic workers have deviated from these scales, it is important to mention so, and to estimate how large uncertainties/discrepancies are between scales.

Reply: This section will be removed.

- Line 329 to 352: This discussion is only concerned with the EOT interval. What about the 400 kyr, and 2.4 Myr cycle that dominate the Oligocene benthic d13C records?

Reply: This section will be removed.

- Line 353 to 370. What about the drop in CO2 at around 24 Ma, during a 1.2 Myr Obliquity cycle "node". May this drop have preconditioned the system for the (large amplitude of the) OMT.

Reply: this section will be thoroughly reconsidered given the proposed changes by the reviewer.

- Line 377: 16esults. Typo?

Reply: Yes, this will be corrected.

- Figure 5: OMT, MOGI, EOGM: please check ages.

Reply: All ages will be checked.

- Line 647 to 657: Very qualitative language. Warm, warmer, etc. Can we not put some numbers to how warm the Oligocene was?

Reply: This is difficult to do with the current availability of data. But we will replace qualitative with quantitative estimates where possible.

- Line 658 to 672: this section is mixing things up. It starts with a description of high resolution benthic d18O records and compilations. Then merges into a discussion of low-resolution SST data, with a reference to Fig. 5 as the only give-away that the topic has changed (admittedly still temperature, but now surface instead of deep ocean). It then comes back to comparing low-res SST (without mentioning that that is what is being compared) to benthic d18O.

Reply: We agree that this section needs some re-shaping. We will re-asses the section and focus it more clearly about the discrepancy of the late Oligocene warming recorded in the benthic foraminifera oxygen isotopes, which is a missing signal in the temperature records.

- Line 660: "most likely affected by a shift of geographical". As far as I am aware there is no doubt about it. Not sure if it is still meaningful to dig out this splicing issue in Zachos 2001. We are two good compilations further (Zachos 2008 and Westerhold 2020). Also confusing this with a true signal in both the Atlantic and Pacific benthic d18O of the Late Oligocene Warming is unhelpful. Is it not smarter to have a section in the paper that discusses benthic d18O. To me, the striking absence of a LOW signal in SST is something more interesting, but that is not discussed.

Reply: The Reference will be updated for this sentence. Additionally the sentence will be re-written in the reshaping of the section. The sentence about the geographical influence will be removed as it is not fitting with the purpose of the section and thus adds to the mix-up in the section as mentioned by the previous comment.

- Line 667: "obliquity". Not true. Both the 1218 and 1264 (both incorporated in Westerhold et al, 2020) are dominated by eccentricity in their spectral power. Low latitude forcing appears to be dominating global climate (in agreement with theory I would argue). How, power from precession is transferred to eccentricity is a topic of discussion, not included in this review.

Reply: We agree with this correction and will re-edit the variability section of the paragraph and also split it off from the long term trends so the paragraph becomes clearer.

- Line 668 -669: "Additionally, stratigraphic constraints on these records are insufficient to assess if this variability is consistent between sites." This is not true either. In the Westerhold et al., 2020 files, Site 1218 and 1264 are correct on the ~110 kyr time scale of tuning. Testing of orbital variability between sites is thus possible, at least on eccentricity time scales.

Reply: Same as above. This is a mistake in the text and will thus be adjusted in the new version.

- Line 669: are we still talking about benthic d18O?

Reply: This line will be clarified.

---

## Author Response (AR1)

Dear Dr. Soreghan,

We thank you for the opportunity to revise our manuscript. Based on the comments of reviewer #1 and reviewer #2 we have been able to significantly improve the manuscript now entitled "Climate variability, heat distribution and polar amplification in the warm unipolar 'icehouse' of the Oligocene". Below we respond to all comments and indicate how we have adapted the paper based on them.

Sincerely, also on behalf of my coauthors,

Dominique Jenny

Based on reviewer #1 the following changes were made (marked in orange):

Review of "Climate variability, heat distribution and polar amplification in the unipolar 'doubthouse' of the Oligocene"

General comments:

The purpose of the paper is confusing. It wasn't clear from the start that it is a literature review and interpretation. The paper also tries to do too much by reviewing the chronostratigraphy, the geography, model-data comparison, fauna and flora, introducing new data using NRL, etc., etc. This could be mitigated by stating earlier (title, abstract, intro) and more clearly the purpose and structure of the paper and by restructuring sections of the paper (see below). The discussion/conclusion sections are a little disappointing. The discussion section doesn't offer any insights into the mysteries or speculation about the remaining Oligocene mysteries (e.g. model-data mismatch, CO2-climate mismatch). The conclusion section was obviously a last-minute addition and needs to be redone. Critiques aside, I appreciated the thorough overview and learned some new things about the Oligocene. This could be a nice addition with some additional work.

Reply: The sections that aren't directly crucial to understand the data review we present (e.g. chronostratigraphy, fauna and flora) were removed. In line with this, we will also revisited the discussion and re-wrote the conclusion section.

Title:

The title is misleading. This is a review paper and the title should reflect that.

Reply: We changed the title to: "Climate variability, heat distribution and polar amplification in the warm unipolar 'icehouse' of the Oligocene"; thereby also incorporating a comment by Reviewer #2.

0 Abstract:

Page 2, Line 14-15 – I don't understand this sentence. Wouldn't the Oligocene be a good analogue for a future climate state with unipolar glaciation?

Reply: We restructured the abstract to make it clearer.

Page 2, Line 28 – Delete "while still maintaining a unipolar icehouse state."

Reply: This section was removed

1 Introduction:

Most of the paper is well written. The Introduction is the exception and requires careful editing.

Page 3, Line 4 – "equilibrate" is used incorrectly here.

Reply: The wording was changed

Page 3, Line 43-44 - Awkward sentence. Please fix.

Reply: Sentence was fixed

Page 3, Line 51-52 - Awkward phrasing. Please fix.

Reply: Sentence was fixed

Page 4, Line 80 - I'm not sure why whales are mentioned here.

Reply: The sentence here was adjusted to reflect the importance of such an evolution in the context of the sentence. (see lines 78-81)

Page 5, Figure 1 caption - Maybe the colors didn't transfer correctly, but some of the colors and shapes mentioned in the caption don't match the colors in the legend of the figure (for (b)). Misspelling of "Transition" as well.

Reply: The figure caption was corrected and the colours were adapted to be correctly representing the methods.

Page 6, Line 94 – The authors mention variability in continental ice volume paced by eccentricity and obliquity. The authors should follow that with an explanation of why/if that's significant and how it might be seen in the proxy records (given that they are low resolution).
Reply: An explanation was added here about the significance of the cyclicity of the record, and where it can be seen. (see lines 83-93)

Page 6, Line 99 – "Here.." Should be start of a new paragraph.
Reply: A new paragraph was made here.

Page 6, Line 100 – "after a chronostratigraphic section" Awkward phrasing.
Reply: The section between line 94 and 102 was extended and rephrased to include more detail on the paper's purpose, also considering the above remark entitled 'General Comments' by the reviewer.

Page 6, Lines 99-104 - At the end of the introduction, the authors finally explain that the purpose of this paper is to review the current state of knowledge regarding the Oligocene climate and thus provide constraints on boundary conditions, compile proxy data, and identify points of interest. This point should be made much sooner so that readers understand the purpose of the paper.
Reply: We specify in the abstract This was considered in the re-shaping of the introduction.

Page 6, Lines 99-104 - There is no mention of numerical climate simulations as in the abstract. Model-data comparisons are made later in the paper, and should be mentioned here. There is also no mention of constraining ocean circulation or ice dynamics though a large part of the introduction discusses these. The authors need to make clearer how climate simulations are used in this paper with some quick details on them, if anything on the ocean or ice state of the Oligocene is explored further, and potentially more details on the proxy records used and for what reason? There is not much on the purpose of the paper and it's all at the very end of the introduction - they could expand on what the paper will discuss and could mention the main purpose it serves sooner in the introduction.
Reply: Based on the reviewer's comments, we have reorganized the introduction so that the order of presentation now follows: 1) the Oligocene is a valuable climate state to study, 2) a very basic description of general climatologically-relevant aspects of the Oligocene in that context, and 3) the purpose of this study: a review to provide a starting point regarding the state of paleoclimate data and its comparison to current generation fully-coupled climate model simulations.

2 Oligocene chronostratigraphy and event nomenclature:
This section should be removed completely from the paper or moved to Supplemental Data.
Reply: Most of was removed following the comments by Reviewer #2 and only the sections about relevant nomenclature and isotopic stages as kept to provide the reader with the necessary context (see lines 115-131).

Page 7, Line 112 – "GGSP" should be "GSSP", right?
Reply: This typo was corrected

Page 7, Line 116 - The OMB is stated as 23.04 Ma here and 23.03 Ma in the rest of the sections
Reply: This mistake in the text was corrected.

Page 8, Line 134 – Spell out 'GPTS' as 'geomagnetic polarity time scale (GPTS)'
Reply: This was added.

Page 9, Line 177 - Why does the last zone end at 23.07 Ma when the Oligocene ends at 23.03 or 23.04 Ma?
Reply: This section as removed.

Page 10, Line 196, 198, 201 - Why is there a question mark after 'Axoprunum'?
Reply: This section was removed.

3 Boundary conditions for Oligocene climate:

Page 11, Line 236 – "boundary conditions" is a modeling term and inappropriate here. Replace with "geography" or "continental position"
Reply: This terminology was changed

Page 12, Figure 3 caption – Add an elevation scale bar. Also 'NRL' is supposed to be 'NLR'.
Reply: The typo was corrected. Paleoelevation is very poorly constrained for most of the globe during the Oligocene and thus cannot be added here in this figure with any confidence.

Page 13, Line 266 – "strongly affected regional ocean circulation…" Here the authors are repeating a hypothesis that is by no means certain and should equivocate appropriately "may have strongly affected". They do this throughout the paper, stating hypotheses, sometimes model results, as fact or certainty. Please clarify appropriately.
Reply: The phrasing here was adjusted accordingly.

Page 15, Line 348 – The authors mention an increase in ocean methane hydrates, peat, and wetlands as a result of colder temperatures and ice expansion. They could add a sentence that explains how the expansion of those environments are a consequence of colder temperatures and if there is evidence of these expanding during the Oligocene or if it is just a probable guess given the cooling climate.
Reply: This section was removed.

Page 15, Line 358-380 – How does these $CO_2$ levels/trends compare with the new $CO_2$ from Honisch et al., Science?
Reply: The paper of Honisch et al (Cenozoic $CO_2$ Proxy Integration Project (CenCO2PIP) Consortium) was published after submission of the original version of this manuscript and has now been added to the $CO_2$ section.

Page 16, Line 377 - "16esults" typo
Reply: The typo was corrected

4 Climate proxy data:
Page 16, Line 385 - Why did the authors choose fossil plant remains to analyze and why only fossil plant remains (not paleosol carbonates, etc)?
Reply: There are of course many more possible archives to include, some of which were described in the Flora and Fauna section but much of this information is fragmented and poorly dated, and provided qualitative data only, all of which limit usefulness. For this reason, we focused on vegetation data, which we considered useful for data-model comparison.

Page 17, Line 405-407 – The authors mention an absence of high latitude data (no fossil plant remains suitable for their method). How could this absence affect their results, or how much uncertainty does this add when estimating a meridional temperature gradient for the Oligocene?
Reply: This essentially means we cannot independently assess meridional temperature gradients based on vegetation data, which we hence do not do. We include a recommendation to revisit the cited materials from the Antarctic margin for quantitative analyses, to solve this issue (see lines 281-295).

Page 17, Line 415 - One data point in the NH mid-latitudes is significantly different from the other mid-latitude data points - where is this NH data point? What sort of biome is it from?
Reply: Upon reviewing that specific datapoint, the Makum Coal Field, we re-evaluated its paleolatitude at ~25 N, making its climate reconstruction consistent with the other tropical floras. We have therefore removed the comment about it being anomalously warm amongst mid-latitudinal temperature reconstructions.

Page 18, Figure 4 – Use the same y-axis for each plot. As it lies now, the winter temperatures appear the same as the MATs until you see the slight difference in the y-axes. Also, in the figure caption, it states the (b) plot PI values are in brown but they appear gray just like in plot (a).
Reply: We adjusted the y axis for the plot and changed the figure caption to match the color of the figure.

Page 19, Line 460 - Random parenthesis in statement. Also, the authors could add one more sentence on how pH affects the fractionation of oxygen isotopes in this environment (why an overestimation of 1.5C has to be taken into consideration?).
Reply: The parenthesis was removed and added a sentence on pH influences.

Page 20, Figure 5 - This is a nice figure, though the shapes are so small I can't really place many of the sites to a line. They could instead do just three different shapes for each type of SST proxy. There are also some SST sites in Figure 3 that aren't in Figure 5 (i.e. Site 913 and others), if they aren't included in Figure 5 then are they still used / needed in Figure 3? Why weren't some used in Figure 5?
Reply: Figure 2 shows all used data sites for all figures presented. Specifically, 913 was only used in Figure 5, and not included in Figure4 as it did not span the entire Oligocene. To assess long-term trends, we only include Sites that span throughout all of the Oligocene in Figure 4. We made the symbols more readable and included the colour of the Site in the symbols.

Page 21, Figure 6 - This is a great figure, very concise with a lot of information. It's interesting too that SST seems ~constant from about 40S to 40N for the Oligocene and sort of late Eocene as well, but modern SSTs have a narrower curve peaking from about 20S to 20N. They could mention something on this.
Reply: We added a sentence in the results about the flattened temperature gradient between those latitudes during the Oligocene (see lines 378-379).

Page 22, Figure 7 - Why reconstruct driest month precipitation and not wettest month precipitation? Along those lines, why reconstruct winter temperatures and not summer temperatures (Figure 4)? Could just include one sentence stating the benefits of understanding winter temperatures and driest month precipitation.
Reply: We have included the following sentence in the methods section to explain this (see lines 277-281): "
…as plant species distributions are sensitive to these variables and significant differences exist in the analyzed plant groups for these variables (see Supplementary Data). "

Page 22, Line 496 – The authors generally state that the Oligocene MAP values are similar to modern day, but the mid-latitude values (for either hemisphere) seem higher than modern values (Figure 7)?
Reply: We adjusted the text and discussion accordingly.

Pages 23 and 24 – The authors don't mention similarities or differences between the models here at all (HadCM3L vs CESM). According to Figure 8, it seems there are a lot of similarities between how the models reconstruct surface temperature compared to the proxy records. It's interesting that you see such similarities between CESM and HadCM3L (though on that - parts C/F/I of Figure 8 could use a darker color for CESM mean temperature difference since the light pink is a bit difficult to see).
Reply: We added the sentence: "Despite utilizing two very distinct models with different boundary conditions, the temperature discrepancy between the model and data remains similar." at the beginning of this section 4.3.1 Temperature proxy to 515 model comparison, (see lines 415-417)
We changed the colour of the CESM to a darker colour.

Page 24, Line 533 – The authors say the differences plotted are 'calculated as a pointwise difference between the proxy mean value and the model annual mean'. Two thoughts on this: a) how was the model annual mean calculated (was the annual mean for the corresponding grid cell used or the general region / a group of grid cells) and b) are there any seasonal biases within the proxy records that could be causing greater differences since only annual means are used for the simulations? If not, they could just state somewhere that there are no known seasonal biases for these records.
Reply:
The model annual mean value is derived from the nearest grid point to our study site. We added a sentence about that in the paper (see lines 413-415).
All proxies come with potential biases. Vegetation does not actually respond to mean annual temperature but to seasonal extremes. Marine proxies from pelagic organisms may suffer from seasonal growth and export. However, all proxies as used here are calibrated to mean annual temperature with an associated error that principally includes such seasonal biases.

Pages 25 and 26 - Similar to what I said above, similarities and differences between CESM and HadCM3L could be mentioned, I think the light pink color for CESM in Figure 9 parts C/F/I is too difficult to see, and I'm a bit

confused on the 'pointwise difference' method regarding which grid cells were used for the model values. I also think red should correspond with drier (according to data) conditions (negative precipitation difference) and blue should correspond with wetter (according to data) conditions (positive precipitation difference). It's a bit confusing that we are looking at precipitation as a 'data minus model' value where the data usually represents wetter conditions than the model simulates and yet the data points on the plot are in red colors which traditionally represent drier conditions but this is really just a preference.

Reply: The color of the CESM in Figure 8 was adjusted to make it more visible. Also the color for the precipitation was adjusted.

Page 27, Line 583 – See inappropriate capitalizations. "Is of great Importance"
Reply: This section was removed.

Page 27, Section 4.4 – This section on Ice Sheets is out of place. Unlike the rest of Section 4, it has no original data or interpretation. It's simply a literature review and should be included in Section 3.
Reply: This section was removed.

5 Flora and faunal changes:
Similar to last comment, why is this section not with the literature review in section 3?
Reply: This section was removed.

6 Discussion:
Page 31 – The authors mention there's a discrepancy between SH and NH MAP (in Oligocene and not in modern) in the result section (Figure 7). A discussion of why would be welcome here.
Reply: We added that to the precipitation section of the discussion (see lines 514-519).

Page 31, line 686-687 – The authors conclude that the Oligocene was anomalously warm relative to $CO_2$ levels. They should discuss potential explanations for this apparent mismatch.
Reply: We expanded the conclusion section with an outlook with outstanding issues, where the global temperature distribution is discussed as a challenge for future research (see lines 530-562).

7 Conclusions:
It reads as though the authors ran out of energy before the Conclusions. There's no introductory statement and the bulleted list is very sparse. Besides the abstract, the conclusions are the most frequently read section of a paper. The authors don't state that the manuscript is a review or distinguish between their contributions and what's review.
Reply: We rewrote the conclusion section to reflect the main points derived in our review.

Page 31, Line 704 - The second conclusion is specific to the proxy records, not so much what the simulations show, so maybe mention that, though I guess they have a conclusion point specific to the simulations below.
Reply: We rewrote the conclusion and made this clearer.

Page 31, Line 706 - I generally agree with the third conclusion, except Oligocene MAP values around the mid-latitudes appear higher than modern MAP mid-latitude values according to Figure 7.
Reply: We rewrote the conclusion and added this.

Page 31, Line 711 - In line with the last conclusion point, the authors could mention that that may be related to the lack/variability/uncertainty in Oligocene $CO_2$ records. $CO_2$ is not well constrained so perhaps the Oligocene is, in part, not the "icehouse" it seemed to be before because $CO_2$ was, in reality, a bit higher than some older records estimated.
Reply: We re-wrote the conclusion and added this.

Based on reviewer #2 we made the following changes:

Jenny et al present a review of the Oligocene, an 11-million-year long time interval. The review is concerned with:

(1) a general introduction,
(2) Chronostratigraphy and event nomenclature,
(2.1) Planktonic foraminifera zones,
(2.2) Calcareous nannofossil zones,
(2.3) Radiolarian biostratigraphic zones,
(2.4) Dinoflagellate cyst zones,
(3) Boundary conditions for Oligocene climate,
(3.1) Geographical boundary Conditions,
(3.2) Ocean circulation,
(3.4) Carbon cycle,
(4) Climate Proxy data,
(4.1) Temperature,
(4.1.1) Continental Mean Annual Temperature,
(4.1.2) Sea Surface Temperature,
(4.2) Precipitation,
(4.3) Data-model comparison,
(4.3.1) Temperature proxy to model comparison,
(4.3.2) Precipitation proxy to model comparison,
(4.4) Ice sheets,
(5) Flora and Faunal changes,
(6) Discussion,
(Note, 6.1 is missing).
Reply: Numbering was corrected

(6.2) Temperature trends and variability,
(6.3) Precipitation, and
(7) Conclusions.

This review constitutes a very ambitious, admirable, and important project, that necessarily needs to make broad-brush summaries and interpretations about the main climatic and oceanographic factors that determined the Oligocene palaeoclimate system. In principle, I am sympathetic to this effort. The paper is well written. However, I believe that for this paper to become an authoritative reference paper much more work is needed. I am no expert on all topics reviewed here. But on topics on which I do consider myself reasonably knowledgeable I find that they are often marked by too many inaccuracies, some outdated discussion points, and several key omissions. By proxy, I have therefore little faith that the parts of the review concerned with topics on which I am no expert, accurately capture the current state of knowledge. I am left with the impression that this paper 'could have been so much better'.

To me the paper suffers from not having included enough Oligocene workers, many of whom have dedicated their professional lives to understanding topics ranging from age models, marine microfossils (calcareous nannofossils, foraminifera, radiolaria, etc.), macrofossils (animals and plants), ice sheet modelling, palaeogeography, etc. Compare, for example, this Oligocene review to Steinthorsdottir et al., 2021, who consulted 3 to 4 times as many experts for their Miocene review. (Here my criticism is mainly directed at coauthors Bijl, Huber, and Sluijs; all experienced and senior workers, some of whom have been involved in the Steinthorsdottir effort. To me, and as far as I can judge, it seems that with greater input from these authors, also in study design, the manuscript could have been lifted to a higher level already. Jenny's efforts in leading this study, as a PhD student, is admirable to say the least.)

I believe that revising this manuscript will result in a much better paper. To me the authors have two options: Option 1) shorten the manuscript considerably by removing much of the review text that is not pertinent to the temperature/precipitation compilation and model comparison, i.e. what I consider the main strength of this paper, and expertise of the authors, or Option 2) include more expert knowledge, work with the text and develop the discussion to lift this paper to a higher level. I recommend the second option, and I believe it could

make, after substantial revisions, for an authoritative Oligocene reference paper. Ultimately, it is the author's choice.

I will provide some below and wish the authors much success with revising this document and planning the next steps ahead. I am sorry that I am not more positive in this instance.

Reply: We agree and chose for the reviewer's Option 1. We shortened the manuscript to the sections that are only important to understanding the newly presented compilation of temperature and precipitation data, and data-modelling comparison and only retain the background information that is important for their context. Thus Chapter 2, large parts of Chapter 3, section 4.4 and Chapter 5 were removed. Relevant information was included in the introduction or discussion chapters. This makes a much more concise paper, focused on the main issue identified in the reformulated (also considering the comments by Reviewer #1) introduction.

Major comments

Structure of the paper: I believe that inclusion of subheadings concerned with magnetostratigraphy, astrochronology, oxygen and carbon isotope stratigraphy would strengthen the paper. Perhaps compare to Steinthorsdottir et al., 2021, to see how they went about their Miocene review. If this comparison is unjustified, please say so in the rebuttal.

Reply: Sections 2.1-2.4 were removed from the paper.

Doubthouse/Unipolar Doubthouse: Most workers use "Doubthouse" to describe the latest Eocene (between the MECO and EOT), where the climate system was cooling and "doubting" to jump from a "Greenhouse" state to a "Unipolar Icehouse" state. I much prefer to reserve the phrase "Doubthouse" for the Late Eocene, and the usage of "Unipolar Icehouse" to describe the state of the Oligocene to Pliocene time interval. As an aside, I also prefer "Unipolar Icehouse" over the phrase "Coolhouse". The usage of "Unipolar Doubthouse" in your title is a mixture of "Unipolar Icehouse" and "Doubthouse" and is confusing to me. I am not entirely sure what is meant by this phrase. Is the system "doubting" to be unipolarly glaciated? Or "doubting" to go back to a "Greenhouse" state, or to a "Bipolar Icehouse" state?

Reply: We altered the title accordingly, as mentioned above to reviewer #1 comments.

Figure 1: This figure has many inaccuracies. Almost all boundaries or events (apart from those around the EOT, which seem to be OK) are indicted in the wrong place. Furthermore, the MOG is not a previously defined event, the MOGI is. The MOGI as drawn does not cover the MOGI as defined in the literature. The OMT does not cover the largest glaciation across the OMB. The Mi-1 points to an d18O low, not a d18O high, and moreover it is best to abandon Mi and Oi nomenclature completely (more about this below).

Reply: All boundaries were checked and moved where necessary, and we abandoned the Mi and Oi points terminology. Also, MOG was changed to MOGI in the figure as well as text.

Line 108/109, the Oligocene time scale: For an Oligocene review paper, it is important to get the age scale right, and to describe it explicitly (not just mentioning it in a figure caption). The absolute ages of the polarity time scale in the GTS2012 (i.e., the GPTS2012) are of greater quality than those incorporated in the GTS2020/GPTS2020, especially for the latest Oligocene where the GTS2020 incorporated erroneous ages. Admittedly, large efforts have been put toward improving biostratigraphic markers and zonations for the GTS2020, but also their absolute ages are unfortunately too often based on an inferior GPTS2020 scale (compared to the GPTS2012). I recommend using GPTS2012 to any Oligocene (age model) worker, and an Oligocene review paper needs to address the errors in the G(P)TS2020, if the authors decide to stick with this age scale as their preferred option.

Reply: For our contribution regarding large-scale Oligocene trends and data-model comparisons of larger time intervals, the choice of time scale is relatively unimportant. We have therefore chosen to retain the 2020 time scale. We included this information in the methods section (see lines 275-277) and in the data file so that this can always be updated when an improved time scale is available.

Line 353 to 380: This CO2 discussion needs more work. What are the implications for radiative forcing? For example, how to explain the Late Oligocene Warming whilst CO2 is decreasing? This is a major topic in Oligocene research, and it is not mentioned or discussed. For this manuscript to become an authoritative review, such a major climatic enigma needs to be addressed, because it will help with formulating new and important research questions.

Reply: We included a section 6.3 on outstanding issues and future research, including the presumed late Oligocene warming and its potential discrepancy with the CO2 reconstructions. Additionally, we discussed it in

the light of the recently published CO2 record by the Cenozoic CO2 Proxy Integration Project (CenCO2PIP) Consortium.

Discussion and Conclusions, both seem quite superficial to me. Can we take a step back, and distil major research questions for the Oligocene?
Reply: The discussion was rewritten to be more pinpointed at the questions posed in the introduction. The conclusion was re-written entirely and we added a section on outstanding issues.

Minor comments
Line 13: Perhaps start the abstract with why the Oligocene is of any interest to anybody? It may help draw some more readers in.
Reply: We changed the first sentences of the abstract.

Line 14: Not entirely sure why this would lead from the first line of the abstract. To me the gradients could be a good analogue. Ice sheet geometry, palaeogeography, etc, seem to me the limiting factors in the Oligocene being a palaeo analogy for long term future climate states.
Reply: We rephrased those sentences (see lines 13-15)

Line 23: proxy data instead of proxy based data?
Reply: This was changed to proxy data (see line 21)

Line 25: "In line with previous…" Does this refer to Oligocene modelling efforts? Or modelling of warmer climates in the general. I thought that this was a feature of Eocene modelling output but was unaware of this being a problem in general, also for the Oligocene unipolar icehouse. Perhaps clarify?
Reply: We clarified this sentence that we indeed mean previous proxy-model comparisons. (see lines 23-26)

Line 23 to 27: should we learn a lesson from these observations? Is it worth making that lesson explicit in the abstract (e.g., much more effort needs to be invested in improving Oligocene climate simulations?)
Reply: We added a final conclusional sentence to the abstract about the model simulations. (see lines 23-26)

1: Position Mi-1 is wrong. In general, it is best to refrain from using Oi and Mi numbering. Oi and Mi numbers were originally defined (in very low-resolution records from the 1980s and early 1990s) as oxygen isotope zones, like biozones, that lasted several hundreds of thousands to millions of years (see original Miller papers and Wright papers). Over the years, these zone numbers were used to refer to the oxygen isotope "highs" (often called "events") at the base of the original oxygen isotope zones, yet this is not how they were originally defined. The arrows in Fig 1, suggest that the authors also use the Oi and Mi numbers to indicate events. If this usage is adopted, which I do not recommend, then please refer to high oxygen isotope values. E.g., the position of the Mi-1 is not correct in this figure (you refer to an oxygen isotope low). Furthermore, even when the Mi-1 (short for "Miocene oxygen isotope zone 1") is indicated correctly in Fig. 1, it would still fall in the latest Oligocene (yet another reason to abandon Oi and Mi numbers). Lastly, nobody has even been able to explain to me what makes the Oi1b, Oi2a, and Oi2b stand out as particularly interesting, or exceptional "events", probably because these numbers were devised as zones originally.
Reply: We removed all Mi and Oi arrows/ events from the figure(s).

1: MOGI instead of MOG? Also, this band is not drawn according to the original definition. You may choose to redefine the main intervals of interest, but then they need to be defined in the text.
Reply: We do define the MOGI in section 2 but only with the Chrons, hence we added the actual ages of those Chrons to the text (see lines 142-144). We also rename the MOG in the figure 1 to MOGI and made sure that in the text it is always called MOGI.

Line 90: Oligoene (missing c)
Reply: The typo was corrected.

Line 90: MOG, where G stands for "glacial". In MOGI, this stands for Glacial Interval, and refers to a longer period of mean elevated d18O values (punctuated by shorter lasting d18O decreases of about 1 per mil amplitude, interpreted to be Antarctic "interglacials" and/or deep-water warming events). The use of glacial in MOG is a bit strange and makes me think of something more similar to the last Pleistocene glacial cycle.

Reply: We use MOGI in the paper as previously mentioned.

Line 99: See Major Comment #1
Reply: We added subheaders within section 2, to make the section more structured for the reader. Also figure 2 was removed together with sections 2.1-2.4.

Line 109: I strongly recommend using G(P)TS2012 over G(P)TS2020. See Major Comment #3.
Reply: We replied to this comment at the above Major Comment #3.

Line 115: Beddow et al., 2018 did not work on the OM Boundary type section (Lemme Carrosio) but derived an astronomically calibrated age for the base of C6Cn.2n from Pacific sediments (perhaps make that clearer, because the current sentence can be interpreted as if Beddow et al did work on that section). Also, the 100 kyr error is not mentioned by Beddow et al. and seems too large.
Reply: This was corrected.

Line 110 to 125: This section would be much strengthened with the addition of a description of astrochronozones and potentially astro-unit-stratotypes. GSSPs are useful to an extent, but have their limitations (see e.g., Hilgen et al., 2020, Newsl. On stratigraphy). Potentially add a magnetostratigraphic description as well
Reply: Based on above Major Comment 0 and the advice of the reviewer, this section was removed

Lines 126 to 145: Overall a good summary, however, as stated above, I would shy away from reiterating Oi and Mi numbers. EOT, MOGI, and OMT are more useful terms in my opinion. Potentially in addition with the Late Oligocene Warming (LOW). MOGI definition used here, instead of MOG in figure caption. Please note that the MOGI is defined as "a generally cold but highly unstable mid-Oligocene time interval (~0 My to 26.3 My ago), which we refer to as the Mid-Oligocene Glacial Interval (MOGI)", not as "a ~1 Ma year phase of profound cooling/glacial expansion". Usage of Ma is erroneous in this context too (Ma refers to an age not a duration). In general, this section needs some work/detailed checking, as I have not checked all statements, and I may have overlooked other potential errors.
Reply: The figure text was adjusted as mentioned above. In this section the definition of the Mi/Oi was left in as they have often been used in the literature and thus need mentioning in the text but we also explain that this terminology has been abandoned (see lines 105-151). We included the description of MOGI as indicated by the reviewer and use ages from Liebrand et al. and corrected Ma to Myr.

Section 2.1. The GTS2020 is abandoned here?
Reply: This section was removed.

Section 2.2. Similar comment. Agnini et al., 2014, cannot have used GTS2020. Please mention on which age scale these bioevents are given.
Reply: This section was removed.

Section 2.3. Same here.
Reply: This section was removed.

Section 2.4. and here. For Chapter 2, in general, best to refer to an accepted general age scale. I recommend G(P)TS2012. Where biostratigraphic workers have deviated from these scales, it is important to mention so, and to estimate how large uncertainties/discrepancies are between scales.
Reply: This section was removed.

Line 329 to 352: This discussion is only concerned with the EOT interval. What about the 400 kyr, and 2.4 Myr cycle that dominate the Oligocene benthic d13C records?
Reply: This section was removed.

Line 353 to 370. What about the drop in CO2 at around 24 Ma, during a 1.2 Myr Obliquity cycle "node". May this drop have preconditioned the system for the (large amplitude of the) OMT.
Reply: This section was thoroughly reconsidered given the proposed changes by the reviewer (see lines 240-268).

Line 377: 16esults. Typo?
Reply: This was corrected.

Figure 5: OMT, MOGI, EOGM: please check ages.
Reply: All ages here are averaged for the studied time periods, now explained in the caption.

Line 647 to 657: Very qualitative language. Warm, warmer, etc. Can we not put some numbers to how warm the Oligocene was?
Reply: This is difficult to do with the current availability of data. But we have replaced qualitative with quantitative estimates where possible.

Line 658 to 672: this section is mixing things up. It starts with a description of high resolution benthic d18O records and compilations. Then merges into a discussion of low-resolution SST data, with a reference to Fig. 5 as the only give-away that the topic has changed (admittedly still temperature, but now surface instead of deep ocean). It then comes back to comparing low-res SST (without mentioning that that is what is being compared) to benthic d18O.
Reply: We re-shaped this section and focused it on the discrepancy with the late Oligocene warming recorded in the benthic foraminifer oxygen isotopes, which is a missing signal in the SST records (see lines 470-477).

Line 660: "most likely affected by a shift of geographical". As far as I am aware there is no doubt about it. Not sure if it is still meaningful to dig out this splicing issue in Zachos 2001. We are two good compilations further (Zachos 2008 and Westerhold 2020). Also confusing this with a true signal in both the Atlantic and Pacific benthic d18O of the Late Oligocene Warming is unhelpful. Is it not smarter to have a section in the paper that discusses benthic d18O. To me, the striking absence of a LOW signal in SST is something more interesting, but that is not discussed.
Reply: The reference was updated for this sentence. Additionally, the sentence was re-written in the reshaping of the section (see lines 477-485). The sentence about the geographical influence was removed.

Line 667: "obliquity". Not true. Both the 1218 and 1264 (both incorporated in Westerhold et al, 2020) are dominated by eccentricity in their spectral power. Low latitude forcing appears to be dominating global climate (in agreement with theory I would argue). How, power from precession is transferred to eccentricity is a topic of discussion, not included in this review.
Reply: This section was rewritten (see lines 477-562).

Line 668 -669: "Additionally, stratigraphic constraints on these records are insufficient to assess if this variability is consistent between sites." This is not true either. In the Westerhold et al., 2020 files, Site 1218 and 1264 are correct on the ~110 kyr time scale of tuning. Testing of orbital variability between sites is thus possible, at least on eccentricity time scales.
Reply: Same as above. This is a mistake in the text and was thus adjusted in the new version (see lines 477-562).

Line 669: are we still talking about benthic d18O?
Reply: This was clarified (see lines 530-560).